# Molecular Sentinels: Unveiling the Role of Sirtuins in Prostate Cancer Progression

**DOI:** 10.3390/ijms26010183

**Published:** 2024-12-28

**Authors:** Surbhi Chouhan, Naoshad Muhammad, Darksha Usmani, Tabish H. Khan, Anil Kumar

**Affiliations:** 1Lyda Hill Department of Bioinformatics, University of Texas Southwestern Medical Center, Dallas, TX 75235, USA; 2Cecil H and Ida Green Center for Systems Biology, University of Texas Southwestern Medical Center, Dallas, TX 75235, USA; 3Department of Radiation Oncology, School of Medicine, Washington University, St. Louis, MO 63130, USA; 4Department of Ophthalmology, Washington University School of Medicine, St. Louis, MO 63130, USA; 5Department of Pathology and Immunology, Washington University School of Medicine, St. Louis, MO 63130, USA; 6Department of Systems Biology, Beckman Research Institute of City of Hope, Monrovia, CA 91016, USA

**Keywords:** prostate cancer (PCa), prostatic intraepithelial neoplasia (PINs), sirtuins, histone deacetylase, Janus-faced sirtuin, cancer signaling, immune modulation, chemoresistance, targeted chemotherapy

## Abstract

Prostate cancer (PCa) remains a critical global health challenge, with high mortality rates and significant heterogeneity, particularly in advanced stages. While early-stage PCa is often manageable with conventional treatments, metastatic PCa is notoriously resistant, highlighting an urgent need for precise biomarkers and innovative therapeutic strategies. This review focuses on the dualistic roles of sirtuins, a family of NAD+-dependent histone deacetylases, dissecting their unique contributions to tumor suppression or progression in PCa depending on the cellular context. It reveals their multifaceted impact on hallmark cancer processes, including sustaining proliferative signaling, evading growth suppressors, activating invasion and metastasis, resisting cell death, inducing angiogenesis, and enabling replicative immortality. SIRT1, for example, fosters chemoresistance and castration-resistant prostate cancer through metabolic reprogramming, immune modulation, androgen receptor signaling, and enhanced DNA repair. SIRT3 and SIRT4 suppress oncogenic pathways by regulating cancer metabolism, while SIRT2 and SIRT6 influence tumor aggressiveness and androgen receptor sensitivity, with SIRT6 promoting metastatic potential. Notably, SIRT5 oscillates between oncogenic and tumor-suppressive roles by regulating key metabolic enzymes; whereas, SIRT7 drives PCa proliferation and metabolic stress adaptation through its chromatin and nucleolar regulatory functions. Furthermore, we provide a comprehensive summary of the roles of individual sirtuins, highlighting their potential as biomarkers in PCa and exploring their therapeutic implications. By examining each of these specific mechanisms through which sirtuins impact PCa, this review underscores the potential of sirtuin modulation to address gaps in managing advanced PCa. Understanding sirtuins’ regulatory effects could redefine therapeutic approaches, promoting precision strategies that enhance treatment efficacy and improve outcomes for patients with aggressive disease.

## 1. Introduction

Prostate cancer (PCa) stands as a major global health challenge, ranking as the fifth leading cause of cancer associated death worldwide [1,2]. As a highly heterogeneous tumor, PCa progresses through a sequence of well-defined stages, including prostate intraepithelial neoplasia (PIN), invasive carcinoma, and ultimately, hormone dependent or independent metastasis [3]. The diagnosis of PCa involves digital rectal examination, prostate-specific antigen (PSA) analysis, prostate biopsies, and advanced imaging techniques, like multiparametric MRI (mpMRI) and PSMA-PET/CT, which enhance precision [4]. Treatment for localized PCa includes active surveillance, radiotherapy, and radical prostatectomy; whereas, advanced disease management relies on androgen deprivation therapy (ADT), chemotherapy, immunotherapy, and novel hormonal agents [5]. Although early-stage PCa is effectively managed through surgical and radiation-based approaches, metastatic PCa remains largely incurable [6]. Also, combination therapies are increasingly effective, yet prostate cancer remains incurable, prompting research into nanotechnologies, gene therapy, and alternative approaches [7]. Moreover, advances in biomarkers and imaging enhance diagnosis, risk stratification, and personalized management, highlighting the need for deeper insights into disease mechanisms and novel therapeutic targets.

At the molecular level, mutations, epigenetic modifications, metabolic alterations, and the dysregulation of signaling pathways drive the transformation of normal prostate cells into cancerous ones, promoting uncontrolled proliferation, survival, and metastasis, thereby mirroring the fundamental mechanisms of oncogenesis observed in other cancers [8,9,10,11,12,13]. Among various types of epigenetic alterations, histone acetylation plays a pivotal role in PCa reprogramming. Histone acetyltransferases (HATs) and deacetylases (HDACs) are central regulators of these modifications, modulating protein acetylation patterns that drive cancer progression [14,15,16]. Sirtuins, a family of NAD+-dependent deacetylases (SIRT1-7), play diverse roles in tumor progression, with certain isoforms acting as tumor suppressors, while others promote tumorigenesis, positioning them as potential targets for cancer therapy. Accumulating evidence highlights sirtuins as critical upstream modulators of cancer signaling cascades, influencing tumor growth, metabolic regulation, genomic stability, and treatment resistance [17]. Among cancer types, each sirtuin exhibits distinct roles as tumor suppressors or oncogenes, influenced by their localization, substrates, and even the stage of cancer [18,19,20]. SIRT1 in general promotes proliferation in cancers, including PCa, but suppresses growth in glioblastoma and bladder carcinoma [21,22,23,24,25]. Conversely, SIRT2 regulates the cell cycle and suppresses tumorigenesis in gliomas, while promoting proliferation and disease aggressiveness in prostate cancer [26,27,28,29,30,31]. Mitochondrial sirtuins (SIRT3–SIRT5) regulate metabolism, with SIRT3 and SIRT4 acting generally as tumor suppressors, while SIRT5’s role remains ambiguous across caner types [17,19,32,33,34,35,36,37,38,39,40]. SIRT6 is mostly reported to enhance genomic stability and counteracts metabolic reprogramming [22,23,24,25,41,42]. SIRT7, often oncogenic, drives ribosomal RNA synthesis and correlates with poor prognosis [19,43,44,45,46,47,48,49,50,51,52,53]. Collectively, sirtuins govern critical processes, including cellular metabolism, chromosomal stability, and gene expression, highlighting their complex, context-dependent roles in cancers. Thus, gaining a deeper understanding of sirtuins in cancers, specifically in PCa, is essential.

This review aims to elucidate the multifaceted roles of sirtuins in PCa, emphasizing their dualistic functions as both oncogenic drivers and tumor suppressors. Specifically, SIRT1, SIRT2, SIRT6, and SIRT7 are implicated in promoting PCa progression, while SIRT5 displays a paradoxical role, simultaneously aiding DNA repair and inhibiting apoptosis, yet fostering tumor growth through androgen receptor (AR) interactions. In contrast, SIRT3 and SIRT4 primarily function as tumor suppressors by regulating cellular metabolism, cell cycle progression, and chromosomal stability, thereby counteracting tumorigenesis. By clarifying these intricate and often opposing roles, this review highlights an urgent need to deepen our understanding of how individual sirtuins contribute to PCa pathogenesis. Such insights are crucial for developing precision therapeutic strategies that leverage the complex nature of sirtuins, ultimately enabling the more effective modulation of their activity to improve outcomes for patients with PCa.

## 2. Structural Variations Among Sirtuins and Their Functional Implications

The structural architecture of sirtuins, though unified by a conserved NAD^+^-binding catalytic core, exhibits notable diversity that underpins their distinct functional roles in cellular processes [54]. At the heart of all sirtuins lies the Rossmann fold, a universal feature essential for NAD^+^ binding and enzymatic catalysis [55]. This fold is anchored by critical histidine residues, such as His363 in SIRT1 and His187 in SIRT2, that orchestrate their NAD^+^-dependent deacetylase or mono-ADP-ribosyltransferase activities [56]. An additional layer of structural stability is provided by zinc-binding motifs, consisting of conserved cysteines [57,58] (as summarized in Table 1), which not only ensure structural integrity but also serve as a scaffold, maintaining the precise conformation of the catalytic domain required for enzymatic activity [59,60].

Despite conserved features, variations in the N- and C-terminus, including differences in sequence and length, significantly affect sirtuins’ interactions with substrates, cellular localization, and functional diversity [61,62,63,64,65]. Nuclear sirtuins, like SIRT1, SIRT6, and SIRT7, have longer terminal regions that enable interactions with chromatin and DNA repair proteins, supporting gene expression and genomic stability. In contrast, mitochondrial sirtuins, such as SIRT3, SIRT4, and SIRT5, have shorter termini, reflecting their roles in mitochondrial metabolism [51,66,67,68,69]. This structural divergence, combined with variations in substrate-binding pockets, endows each sirtuin with a distinct functional role. For instance, SIRT1 has a broad substrate range, including histones and p53, enabling it to regulate apoptosis and metabolism, while SIRT6 exhibits specificity for histones, playing a key role in chromatin remodeling [70,71,72,73,74,75]. Structural variations among sirtuins also extend to their localization signals, which dictate their specific subcellular destinations and, thus, their functional roles. Nuclear localization signals in sirtuins, like SIRT1, SIRT2, and SIRT6, enable nuclear functions, such as transcriptional regulation and DNA repair, with SIRT6 specifically targeting chromatin for telomere maintenance. In contrast, mitochondrial targeting sequences in SIRT3, SIRT4, and SIRT5 direct them to mitochondria, where they regulate metabolic enzymes [64,67,76,77,78,79,80]. Variations in substrate-binding sites further contribute to functional diversity, with SIRT1 binding a broad range of substrates, while SIRT6 has a more selective binding pocket for histones, supporting its specialized role in chromatin regulation [81,82,83,84]. While exhibiting structural and functional diversity, sirtuins play a central role in maintaining cellular homeostasis, particularly in metabolism, gene expression, and longevity [44,85,86,87,88]. These differences also underscore the complexity of sirtuin regulation across cancers from different tissues and highlight the importance of their structural variations in understanding their roles in tissue-specific carcinogenesis. The individual domains of sirtuins and their structural variations are outlined in Figure 1.

## 3. Sirtuins and Prostate Carcinogenesis

The sirtuin family has emerged as a fascinating regulator of PCa, orchestrating a complex network of cellular processes that are crucial for tumor development, progression, and therapeutic resistance. These NAD+-dependent deacetylases are far more than metabolic sensors. They are key modulators of autophagy, oxidative stress response, epigenetic regulation, and metabolic adaptation. However, what makes sirtuins particularly intriguing is their duality, i.e., acting as tumor suppressors in some contexts, while promoting tumorigenesis in others. This Janus-faced nature reflects their ability to adapt to cellular demands and environmental stressors, making them both guardians and facilitators of cancer cell survival. For instance, SIRT1, often acknowledged for its role in maintaining mitochondrial health and reducing oxidative stress, paradoxically contributes to chemoresistance and castration-resistant prostate cancer (CRPC) by fine-tuning AR signaling and enhancing DNA repair [22,89,90]. Meanwhile, SIRT3 and SIRT4 act as metabolic gatekeepers, repressing oncogenic signaling, preventing epithelial mesenchymal transition (EMT) and disrupting cancer cell metabolism, while SIRT2 and SIRT6 shape the tumor landscape by influencing AR sensitivity and tumor aggressiveness, with SIRT6 enhancing cell survival and metastatic potential in advanced disease [91,92,93,94]. Even more fascinating is SIRT5’s Janus-faced role, toggling between promoting and suppressing oncogenic pathways by regulating key metabolic enzymes and signaling cascades [95,96,97]; SIRT7 completes this complex puzzle by supporting cancer cell proliferation and adaptation to metabolic stress via its nucleolar and chromatin regulatory functions [98]. Together, this evidence presents a dynamic picture of sirtuins as versatile regulators of prostate cancer, capable of both fueling and suppressing tumor progression depending on the context (Figure 2). The following sections will delve into the unique and overlapping functions of individual sirtuins in prostate cancer, unraveling their potential as critical determinants of disease behavior and clinical outcomes.

### 3.1. Tumor Promoter Sirtuins

#### 3.1.1. SIRT1

SIRT1 is one of the most extensively studied sirtuins for its multifaceted role in PCa biology. Its involvement spans various critical aspects of tumor progression, making it a pivotal molecule in PCa research. To provide a comprehensive understanding, we will subsection this topic into several key areas. These include the prognostic implications of SIRT1 in PCa, highlighting its potential as a biomarker; its role as a central regulator of metabolism and redox balance, emphasizing its influence on tumor energetics and oxidative stress responses; and its involvement in transcriptional and epigenetic regulation, where it modulates gene expression through chromatin remodeling. Additionally, we will explore SIRT1’s interaction with AR signaling and its role as a mediator of immune responses, particularly in fostering an immune-tolerant microenvironment. SIRT1’s regulation of EMT and metastasis will also be discussed, shedding light on its impact on invasion and metastatic potential. Furthermore, we will examine its interaction with other signaling cascades and its regulatory role concerning microRNAs, elucidating how SIRT1 orchestrates complex molecular networks that drive PCa progression. Each of these subsections will delve into the intricate roles of SIRT1, providing a detailed framework for understanding its significance in PCa.

##### Prognostic Implications of SIRT1 in Prostate Cancer

SIRT1 plays a pivotal role in PCa progression, particularly in advanced stages. Early studies using the TRAMP mouse model showed stable SIRT1 expression during early carcinogenesis but a significant increase in poorly differentiated adenocarcinomas, correlating with reduced levels of hypermethylated in cancer-1 (*HIC1*), a regulator of SIRT1 [99]. This elevated expression of SIRT1 in malignant cells was further confirmed through immunohistochemical analyses of human prostate tumors. Notably, homozygous deletion of the *SIRT1* gene in mice led to the development of prostatic intraepithelial neoplasia (PINs), associated with impaired autophagy [100]. The study further suggests that SIRT1’s impact on autophagy is most prominent during the crucial stages of autophagosome maturation and completion, highlighting its pivotal role in sustaining cellular homeostasis. Moreover, in an effort to identify biomarkers of recurrence, a global RNA sequencing analysis was conducted on 106 formalin-fixed, paraffin-embedded prostatectomy samples from 100 patients across three independent sites, leading to the definition of a novel 24-gene signature panel [101]. Its significant role in these pathways highlights SIRT1’s potential as a biomarker for predicting aggressive PCa.

##### SIRT1 as a Central Regulator of Metabolism and Redox Balance in Prostate Cancer

SIRT1 also influences the effects of metabolic dysregulation on PCa outcomes [102]. One study in this context reveals that hyperglycemia significantly reduces docetaxel-induced apoptosis in PCa cell lines, which is intricately linked to the glucose-induced upregulation of *IGFBP2*, which is further regulated by SIRT1 through histone acetylation at the *IGFBP2* promoter [103]. Interestingly, adipokines, such as leptin and resistin, play a significant role in prostate cancer by influencing tumor growth, angiogenesis, inflammation, and metastasis, often linking obesity with increased cancer risk and progression [104]. The SIRT1/leptin axis highlights the interplay between metabolic signaling and tumor progression. Leptin treatment upregulates HIF-1α, enhancing PCa cell adhesion and invasion, along with mitochondrial biogenesis and membrane potential stabilization via uncoupling protein 2. Leptin also counteracts the hypoxia-induced downregulation of SIRT1, maintaining elevated levels that stabilize HIF-1α. Additionally, SIRT1 regulates mitochondrial integrity and ROS production, with lower SIRT1 expression in luminal epithelium correlating with poor prognostic outcomes [105]. Lower SIRT1 expression in luminal epithelium correlates with poor prognosis, promoting cancer progression through the increased acetylation of SOD2 and elevated ROS production. Restoring SIRT1 reduces ROS levels and PARK2 (PRKN parkin RBR E3 ubiquitin protein ligase) translocation, highlighting its role in regulating mitochondrial integrity and PCa-associated signaling [106]. SIRT1 facilitates the interaction between PGC1α and ERG, enhancing the expression of antioxidant genes, like *SOD2* and *TXN*, which promote cell survival under metabolic stress [107]. SIRT1 influences neuroendocrine differentiation (NED) in PCa, especially under ADT, where ADT-induced ROS activate SIRT1, promoting NED via the Akt signaling pathway, independent of N-Myc [108]. Another interesting study reports that NKX3.1 exacerbates oxidative stress, leading to increased H2AX phosphorylation and the degradation of tumor suppressors, like AR and p53 [109]. Interestingly, while inflammatory stimuli, such as conditioned media from activated macrophages, elevate SIRT1 expression, direct oxidative stress from H_2_O_2_ treatment does not, emphasizing the unique influence of inflammatory microenvironments on SIRT1 regulation. This mechanism underscores the multifaceted role of SIRT1 in metabolic regulation and tumor progression in PCa.

##### SIRT1 and Androgen Receptor Signaling in Prostate Cancer

The AR, a steroid receptor critical for mediating testosterone and dihydrotestosterone signaling, plays a pivotal role in PCa progression, particularly in CRPC [110]. SIRT1 acts as a corepressor of AR, suppressing androgen-responsive genes by recruiting nuclear receptor corepressors to AR-responsive promoters. This repression contributes to the development of hormone-refractory PCa and resistance to androgen antagonists [111]. In androgen-deprived LNCaP cells, SIRT1-activated FOXO1 regulates vascular endothelial growth factor-C (*VEGFC*) transcription and downregulates the insulin-like growth factor receptor (IGF-IR) pathway. In turn, *VEGFC* facilitates the androgen-independent reactivation of AR and promotes tumor growth [112]. Additionally, SIRT1 deacetylates the coactivator MYST1, enhancing its interaction with AR and NF-κB, which drives aggressive tumor behavior and therapeutic resistance. This mechanism influences histone acetylation and chromatin structure, underscoring SIRT1’s role in PCa adaptability [113]. Interestingly, while SIRT1 inhibition under androgen deprivation induces cellular senescence, enhancing the efficacy of AR blockade and radiation therapy, its silencing in hormone-responsive LNCaP cells mimics AR inhibition, resulting in smaller tumor formation [114]. Furthermore, SIRT1 antagonists promote AR expression and activity, while SIRT1 directly deacetylates AR, impairing its interaction with coactivators and reducing dihydrotestosterone (DHT)-induced PCa growth. Although SIRT1 activation may protect against early high-grade PIN lesions, its role evolves as the disease progresses [115]. The intricate SIRT1/AR axis highlights the duality of SIRT1’s functions in PCa, with its modulation offering both therapeutic challenges and opportunities for advancing treatment strategies [116].

##### SIRT1 as a Mediator of Immune Response in Prostate Cancer

SIRT1 emerges as a pivotal player in orchestrating the immune landscape within the PCa microenvironment, driving immune evasion and tumor progression. One study reveals that glycolytic cancer-associated fibroblasts (CAFs) secrete lactate, inducing SIRT1-dependent deacetylation and the degradation of T-bet in CD4+ T cells, thereby suppressing the Th1 immune response [117]. Concurrently, SIRT1 fosters regulatory T-cell (Treg) polarization via NF-κB and FoxP3, creating an immune-tolerant microenvironment. This cascade amplifies PCa cell invasiveness, mediated by the miR21/TLR8 axis, underscoring SIRT1’s role in immune evasion and tumor aggressiveness. SIRT1 plays an important role in the neuroendocrine differentiation (NED) of PCa cells, particularly in the context of interleukin (IL)-6 stimulation, which drives the progression of aggressive androgen-independent PCa. Recent investigations reveal that IL-6-driven AMPK-SIRT1 activation promotes neuroendocrine marker expression, including βIII tubulin, neuron-specific enolase, and chromogranin A, while activating p38MAPK signaling to advance NED in LNCaP cells [118]. Intriguingly, mesenchymal stem cells (MSCs) engineered to overexpress SIRT1 (MSCs-Sirt1) demonstrate robust antitumor efficacy. These modified MSCs recruit and activate natural killer (NK) cells and macrophages, boosting interferon-gamma (IFN-γ) and CXCL10 levels in the tumor microenvironment [119]. The resulting pro-inflammatory milieu significantly suppresses prostate tumor growth, an effect diminished when IFN-γ and CXCL10 pathways are inhibited, highlighting SIRT1’s role in immune-mediated tumor suppression. Another layer of complexity involves TNF receptor-associated factor 2 (TRAF2), a regulator of tumor progression and resistance to apoptosis in androgen-refractory PCa. TRAF2 upregulation in high-Gleason score PCa correlates with poor outcomes. Its knockdown enhances apoptosis, while downregulating SIRT1 expression in TRAIL-treated cells, indicating a *TRAF2*/SIRT1 axis that supports tumor survival and growth [120]. SIRT1 also regulates the symbiosis between cancer-associated fibroblasts (CAFs) and cancer cells, significantly influencing oncogenic signaling cascades. Lactate uptake from CAFs shifts the NAD+/NADH ratio in PCa cells, leading to SIRT1-mediated activation of PGC-1α, which drives mitochondrial biogenesis and metabolic activity [121]. This mitochondrial exploitation promotes the deregulation of the tricarboxylic acid cycle, the accumulation of oncometabolites, and the altered expression of mitochondrial complexes, further exacerbating superoxide generation. Collectively, these findings establish SIRT1 as an important regulator that integrates immune responses, metabolic adaptations, and therapeutic resistance in PCa.

##### SIRT1 as a Regulator of EMT and Metastasis in Prostate Cancer

SIRT1 has been implicated in the regulation of EMT and is abnormally expressed in PCa cells, suggesting its role in modulating invasion and metastatic capabilities. Silencing the *SIRT1* gene in PC-3 cells using small interfering RNA (siRNA) significantly reduces cell migration and invasion, accompanied by an upregulation of E-cadherin and a downregulation of mesenchymal markers, like N-cadherin and vimentin [122]. This suggests that SIRT1 suppresses cell adhesion, while promoting cellular mobility. SIRT1 also enhances invasiveness by deacetylating and stabilizing matrix metalloproteinase 2 (MMP2) via the proteasomal pathway [123]. Additionally, it interacts with the transcription factor ZEB1, suppressing E-cadherin expression through histone H3 deacetylation, which facilitates the loss of adhesion and the acquisition of migratory capabilities [124]. Interestingly, SIRT1 engages in a reciprocal relationship with the epigenetic regulator, Matrix metalloproteinase 8 (MPP8), which is critical for EMT and malignant progression. This interaction promotes histone deacetylation and H3K9 methylation at E-cadherin promoters, silencing E-cadherin expression [125]. Disrupting either SIRT1 or MPP8 reverses E-cadherin repression and diminishes the EMT phenotype, emphasizing the integral role of the SIRT1/MPP8 axis in shaping the epigenetic framework that governs cancer cell invasion and metastasis.

##### SIRT1-Mediated Transcriptional and Epigenetic Regulation in Prostate Cancer

SIRT1 plays an important role in modulating transcriptional and epigenetic landscapes in PCa, primarily through interactions with chromatin-modifying complexes and transcriptional regulators. A key mechanism involves the PRC4 complex, which includes SIRT1 and Ezh2, a histone–lysine methyltransferase. This complex alters gene expression patterns via NAD^+^-dependent histone deacetylation, favoring tumorigenesis through the reprogramming of epigenetic states [126]. Dysregulated SIRT1 activity also mediates the degradation of H2A.Z, a histone variant implicated in PCa progression, highlighting its impact on chromatin dynamics [127]. The overexpression of SIRT1 leads to the degradation of H2A.Z via the proteasome mediated degradation, while epigenetic modifying drugs, such as trichostatin A, modulate both SIRT1 and H2A.Z expression. SIRT1 also mediates lysine delactylation on Canopy FGF Signaling Regulator 3 (CNPY3), influencing its cellular localization and promoting lysosomal rupture, thereby triggering pyroptosis [128]. At the chromatin level, SIRT1 also promotes DNA repair and resistance to therapies by enhancing KU70-mediated non-homologous end joining. This process, regulated by lysine-specific demethylase 1 (LSD1), maintains an open chromatin state conducive to DNA repair through histone H4K16 acetylation. This mechanism aids tyrosine kinase inhibitor resistance, further complicating therapeutic interventions [129]. Another interesting study indicates that the deacetylation of FOXO3 by these sirtuins promotes its poly-ubiquitination via the E3 ubiquitin ligase Skp2, ultimately leading to reduced FOXO3 protein levels, particularly in malignant PCa cells, where elevated SIRT1 and Skp2 expressions are observed [130]. Moreover, the overexpression of SIRT1 in PCa cells and tissues promote tumorigenesis by inhibiting the activation of FOXO1 protein [131]. Also, SIRT1 directly binds to and deacetylates FOXO1, resulting in the inhibition of its transcriptional activity. This deacetylation process is enhanced by four and a half LIM domains protein 2 (FHL2), which not only promotes SIRT1-FOXO1 interaction but also inhibits FOXO1-mediated apoptosis [132]. The SIRT1/FOXO1 axis is further elucidated in another study, which highlights that SIRT1 overexpression fosters tumorigenesis; whereas, its inhibition triggers divergent responses in cell viability and growth, mediated by the activation of FOXO1 and influenced by the p53 status of the cells [133]. The inhibition of SIRT1 leads to increased senescence in p53-active PC3-p53 cells and enhanced apoptosis in p53-inactive PC3 cells. The interaction between SIRT1 and p53 reported to obtain the modulate epigenetic silencing of the tumor suppressor *HIC1* results in the upregulation of SIRT1, thereby disrupting the p53-dependent signaling cascades that modulate cellular stress responses and survival [134], thereby fostering tumorigenesis.

Furthermore, another study reports that the SIRT1/FOXO3a axis in PCa is influenced by the overexpression of nicotinamide phosphoribosyltransferase (NAMPT), which plays a crucial role in regulating SIRT1 activity through the regeneration of NAD^+^ [135]. Elevated NAMPT levels enhance SIRT1-mediated deacetylation and FOXO3a expression, providing protection against oxidative stress and supporting tumor resilience. In contrast, NAMPT inhibition reduces FOXO3a levels and weakens antioxidant defenses, underscoring the crucial interaction between SIRT1, FOXO, and NAMPT in prostate carcinogenesis. Additionally, SIRT1 interacts with nicotinamide N-methyltransferase (NNMT), with higher NNMT expression in PCa tissues promoting cell viability, invasion, and migration through SIRT1 upregulation. The suppression of SIRT1 by nicotinamide reduces these aggressive behaviors, highlighting NNMT as a key modulator of SIRT1 in PCa progression [136]. Other studies have revealed that NAMPT sustains de novo lipogenesis in PCa cells, with its inhibition significantly reducing fatty acid and phosphatidylcholine synthesis, disrupting essential metabolic processes for tumor growth. These effects are similar to those seen when sirtuins are blocked or SIRT1 is knocked down, highlighting the complex regulation of lipid metabolism in PCa [137]. Collectively, these findings position SIRT1 as a crucial player in the regulatory networks of PCa, influencing both metabolic adaptations and therapeutic responses through intricate signaling cascades. Collectively, these findings indicate that SIRT1 is a key epigenetic modifier and regulator of the transcriptional machinery in prostate cancer.

##### Interaction of SIRT1 with Other Signaling Cascades in Prostate Cancer

SIRT1 actively participates in the intricate signaling networks that drive PCa progression, interacting with key regulatory molecules and pathways. Retinoic acid receptor responder 1 (*RARRES1*), a candidate tumor suppressor gene, has been shown to enhance SIRT1 expression. This interplay bolsters cellular antioxidant defenses by elevating catalase levels and inhibiting mTOR, thereby countering tumor progression [138]. Additionally, *RARRES1* inhibits angiogenesis in endothelial cells, further underscoring its multifaceted role in tumor suppression and its therapeutic potential in both PCa and angiogenesis-related disorders. The mTOR pathway, particularly mTORC1 signaling, plays a critical role in PCa progression by promoting growth, proliferation, and survival, while inhibiting autophagy [139,140]. The dysregulation of mTORC1 contributes to oncogenesis through enhanced protein synthesis and nutrient metabolism. SIRT1 has been reported to interact with mTORC1 and S6K in PCa cells, modulating autophagy and proliferation [141]. Notably, resveratrol intervention in a PTEN knockout mouse model of PCa suppressed mTORC1 activity and upregulated SIRT1, highlighting the therapeutic potential of targeting this axis. Moreover, SIRT1 interacts with the orphan nuclear receptor TLX, a key promoter of tumorigenesis in PCa. TLX suppresses cellular senescence by repressing cyclin-dependent kinase inhibitor 1A (CDKN1A) and activating SIRT1, creating a favorable environment for malignant cell growth [142]. Elevated TLX expression in high-grade tumors further correlates with aggressive cancer behavior. In addition, SIRT1’s interaction with β-catenin links it to glycolytic regulation. Brassinin, a phytoalexin, disrupts this interaction in PCa, leading to the downregulation of glycolytic proteins, such as PKM2 and GLUT1, while promoting ROS-mediated apoptosis through reduced pro-PARP and pro-caspase 3 [143]. Collectively, these findings position SIRT1 as a central mediator in various signaling cascades, influencing tumor progression, metabolism, autophagy, and apoptosis in PCa.

##### SIRT1 and Interaction with microRNAs in Prostate Cancer

An increasing number of studies highlight recent advancements showing that microRNAs and SIRT1, as key regulators of gene expression, exhibit dual roles in cancer development by either suppressing or promoting tumorigenesis through the modulation of diverse oncogenic pathways, with the microRNA/SIRT1 axis playing a crucial role in regulating cancer signaling cascades [144]. One significant interaction involves miR-34a, a tumor-suppressive miRNA regulated by p53, whose expression is markedly reduced in p53-null and p53-mutated PCa cells [106]. The restoration of miR-34a expression leads to decreased SIRT1 levels, inducing cell cycle arrest and enhancing sensitivity to chemotherapeutic agents, like camptothecin. In hormone-refractory PCa, SIRT1 upregulation correlates with reduced miR-34a levels, contributing to drug resistance via pathways involving proteins HuR and Bcl2 [145,146]. This regulatory dynamic underscores SIRT1’s role in key signaling cascades involving HuR and Bcl2, directly contributing to drug resistance. A novel therapeutic approach using the co-delivery of doxorubicin and miR-34a via micellar systems has been shown to synergistically downregulate SIRT1, suppress proliferation in androgen-independent PCa cells, and enhance the efficacy of chemotherapy [147]. This novel approach not only enhances the cellular uptake and nuclear release of the hydrophobic chemotherapeutic agent but also exploits the synergistic effects of miR-34a to amplify the therapeutic efficacy against tumor cells. Similarly, miR-221 and miR-222, which are overexpressed in androgen-independent PCa, negatively regulate SIRT1. Their inhibition results in increased SIRT1 levels, reducing cell proliferation and migration, while promoting apoptosis [148]. On the other hand, miR-34a-5p modulates the SIRT1/TP53 axis, where its overexpression inhibits apoptosis and promotes cell cycle progression [149]. Combination therapy with paclitaxel and rubone upregulates miR-34a, enhancing chemosensitivity in resistant cancer cells by targeting SIRT1 and its downstream effectors [150].

Conversely, miR-204 targets SIRT1 to enhance the chemosensitivity of PCa cells to doxorubicin by promoting p53 acetylation, which in turn, activates pro-apoptotic proteins NOXA and PUMA [151]. Another study links the downregulation of miR-212 to increased SIRT1 activity, facilitating starvation-induced autophagy and angiogenesis. Similarly, the suppression of miR-449a in ERG-positive PCa elevates SIRT1 expression, creating a feedback loop involving ERG, miR-449a, and SIRT1, which drives invasive cancer phenotypes [152]. Similarly, the downregulation of miR-212 correlates with enhanced SIRT1 activity, thereby facilitating starvation-induced autophagy and promoting angiogenesis and cellular senescence in cancer cells [153]. The attenuation of miR-449a, significantly suppressed in ERG-positive PCa tissues, facilitates the invasive phenotype by directly upregulating SIRT1 expression, thereby creating a feedback regulatory loop between ERG, miR-449a, and SIRT1 [154], and SIRT1 suppression results in decreased ERG expression and is linked to the modulation of p53 acetylation. One study reports that SIRT1 interacts with small extracellular vesicle-associated microRNAs, such as miR-6068. Elevated levels of miR-6068 have been observed in PC-3 and CWR-R1ca cell lines, where it promotes aggressive cellular phenotypes by inhibiting the hypermethylated in the cancer 2 (HIC2)/SIRT1 axis [155]. The suppression of miR-6068 correlates with reduced proliferation, colony formation, and migration in CWR-R1ca cells, highlighting its pivotal role in modulating the tumor microenvironment. Conversely, the upregulation of HIC2 is significantly associated with cytoplasmic localization in PCa tissues compared to benign prostatic hyperplasia, implicating a potential feedback mechanism that enhances SIRT1 expression. These studies underscore the role of the miRNA/SIRT1 axis in PCa, by modulating tumor progression chemoresistance and treatment outcomes.

Thus, SIRT1 assumes a multifaceted role in PCa, functioning simultaneously as a tumor promoter and a modulator of critical signaling pathways that contribute to disease progression (summarized in Figure 3). These insights underscore SIRT1’s potential as a therapeutic target, as its inhibition could not only reverse drug resistance but also disrupt essential oncogenic signaling pathways, thereby presenting promising avenues for the treatment of advanced PCa.

#### 3.1.2. SIRT2

Among the many factors involved in the progression of prostate cancer, SIRT2 plays a unique role by regulating histone acetylation and modulating key signaling pathways. Tissue microarray and xenograft models indicate that SIRT2 reduction is a common feature in CRPC, present in 66% of cases, highlighting its role in epigenetic reprogramming and AR signaling resistance [156]. In contrast, SIRT2 expression in primary PCa is associated with a lower incidence of the disease, suggesting a protective effect in earlier stages. The expression of SIRT2 in CRPC leads to unchecked p300 activity, which drives transcriptional programs promoting tumor aggression and resistance to androgen receptor signaling inhibitors (ARSIs) [157]. Reduced SIRT2 correlates with higher-grade cancers, poorer PSA recurrence-free survival, and diminished responses to ARSIs, underscoring its potential as a therapeutic target in tumors characterized by heightened p300 activity and histone hyperacetylation. SIRT2 plays a critical role in the progression of PCa by modulating histone acetylation, particularly at key lysine residues, such as H3K18. In normal and primary PCa tissues, SIRT2 expression is relatively stable, but it is significantly reduced in CRPC, where it correlates with increased acetylation at H3K9, H3K14, and H3K18, as well as elevated p300 autoacetylation [92,158]. This reduction in SIRT2 activity contributes to hyperacetylation, particularly at H3K18, which has been linked to aggressive tumor behavior and therapeutic resistance.

An interesting study reveals that SIRT2 regulates Leukemia inhibitory factor receptor (LIFR) deacetylation at K620 in PCa cells, influencing LIFR homodimerization and activating the 3-phosphoinositide-dependent kinase 1 (PDPK1)/AKT signaling axis [159]. SIRT2 reverses the GCN5-induced acetylation of LIFR at K620, disrupting its homodimerization, PDPK1 activation, and AKT signaling, thereby attenuating tumor growth and positioning LIFR-K620 acetylation as a biomarker and therapeutic target in PCa. Interestingly, SIRT2 plays a critical role especially in aggressive forms like CRPC and neuroendocrine PCa (NEPC). It modulates histone acetylation and regulates key transcription factors, such as FOXO3 [130]. Mechanistically, SIRT2 promotes the deacetylation of FOXO3, facilitating its poly-ubiquitination by the E3 ubiquitin ligase Skp2, which accelerates its proteasomal degradation. This downregulation of FOXO3, a tumor suppressor with roles in cell cycle arrest and apoptosis, contributes to the unchecked proliferation and survival of malignant prostate cells. In CRPC and NEPC, SIRT2 expression is notably upregulated through the activation of the ERK1/2 signaling pathway [91], which is often associated with chemoresistance in cancer by enhancing cellular survival mechanisms, promoting proliferation, and inhibiting apoptosis [160,161]. Furthermore, SIRT2 influences the metabolic profile of cancer cells by inducing the production of lactosylceramide through the upregulation of B4GALT5 beta-1,4-galactosyltransferase 5 (B4GALT5), which enhances the invasive potential of PCa cells. These findings underscore SIRT2’s critical involvement in both the epigenetic modulation of oncogenic pathways and the metabolic adaptation of PCa, positioning it as a promising target for therapeutic intervention in advanced and treatment-resistant forms of the disease. Figure 4 summarizes diverse functions of SIRT2 in PCa.

#### 3.1.3. SIRT6

SIRT6 plays a significant role in regulating DNA repair, telomere maintenance, energy metabolism, and gene expression across cancer types, while it has been found to be overexpressed in prostate tumors compared to normal or para-tumor tissues, as confirmed by gene profiling and tissue microarray studies. A comprehensive study has indicated that SIRT6 is upregulated in PCa cell lines and tissue specimens, suggesting its potential role in the disease’s progression [162]. Quantitative analyses, including RT-PCR, have demonstrated elevated expression levels of SIRT6 correlating with aggressive disease characteristics, including higher Gleason scores and increased nodal metastasis. Loss-of-function assays reveal that SIRT6 promotes cell proliferation, migration, and invasion, with silencing SIRT6 significantly impairing these critical cancer processes. Its upregulation in PCa patients may serve as a marker for disease progression, highlighting its potential as a therapeutic target. Mechanistically, SIRT6 modulates key signaling pathways, particularly the Wnt/β-catenin pathway, which is frequently altered in cancers and plays a central role in driving EMT and tumor metastasis [13,163,164]. The inhibition of SIRT6 has been shown to disrupt this pathway, leading to decreased β-catenin activity and attenuating EMT progression in PCa cells. Another study reveals that silencing SIRT6 in human PCa cells induces sub-G1 phase cell cycle arrest, heightened apoptosis, and increased DNA damage, accompanied by a reduction in BCL2 expression [165]. Furthermore, SIRT6 deficiency leads to decreased cell viability and enhanced sensitivity to chemotherapeutics. Additionally, the interplay between SIRT6 and the innate immune response further complicates its role in PCa. Studies indicate that SIRT6 modulates necroptosis, a form of regulated cell death, by inhibiting RIPK3-mediated pathways [166]. This regulation influences not only tumor cell survival but also the recruitment of immune cells, such as macrophages and neutrophils, positioning SIRT6 as a critical component of the cancer-immune interface.

Moreover, SIRT6’s involvement in cancer cell lineage plasticity, particularly in the context of neuroendocrine differentiation, represents another dimension of its regulatory capabilities, with a study identifying the G protein-coupled receptor, ADORA2A, as a key driver of lineage plasticity through SIRT6-mediated deacetylation processes [167]. This signaling pathway rewires proline metabolism, which subsequently affects histone modifications and transcriptional outputs that favor neuroendocrine characteristics.

Interestingly, a study reveals that the intricate interactions involving SIRT6 extend into broader metabolic contexts, as demonstrated by findings that the transcription factor E2F1 negatively regulates SIRT6, promoting glycolytic activity in PCa cells [168]. This mechanistic insight, where E2F1 binds to the SIRT6 promoter and suppresses its expression, uncovers a critical link between metabolic reprogramming and SIRT6 activity. Finally, the identification of protein interaction networks involving SIRT6, particularly within the PIN7 network, emphasizes its role in age-related diseases and cancer. The interactions between SIRT6 and various proteins implicated in tumor suppression, such as p53 and MYBBP1A, delineate pathways that may regulate crucial processes, such as the cell cycle and apoptosis [169]. Given the significant deletions observed on chromosome 15 in various cancers, including PCa, further investigation into these interactions could unveil novel therapeutic strategies aimed at reinstating SIRT6 function and exploiting its regulatory capabilities to combat tumor growth and metastasis. Collectively, these insights into SIRT6’s multifaceted roles in PCa establish a compelling case for its consideration as a central target in developing innovative treatment modalities for this prevalent malignancy (also outlined in Figure 5).

#### 3.1.4. SIRT7

The investigation of SIRT7 reveals its potential role as a predictive biomarker for aggressive PCa [170], characterized by a lack of reliable prognostic indicators, with immunohistochemical analysis of 57 patients showing significantly elevated SIRT7 expression in tumor tissues compared to adjacent healthy tissues and a positive correlation with cancer grade [98]. Mechanistic studies revealed that silencing SIRT7 reduced the migration of androgen-independent PCa cell lines DU145 and PC3, while its overexpression increased migration and invasion in the less aggressive LNCaP cell line and conferred resistance to docetaxel, highlighting its role in promoting both the aggressiveness and treatment resistance of PCa cells. The exploration of SIRT7’s involvement in metastatic processes reveals its critical role in the epigenetic reprogramming associated with EMT in prostate carcinomas, with the depletion of SIRT7 in both epithelial and mesenchymal cancer cells reversing aggressive traits and underscoring its function as a regulator of metastatic behavior [171]. Eminently, SIRT7’s influence extends to the p53 signaling pathway, highlighting its significance in cancer biology, with the protein–protein interaction network, PIN7, implicated in age-related diseases, including cancer [169]. Pathway enrichment analyses in this study reveal that p53 signaling serves as a predominant mediator of PIN7’s effects, with SIRT7 influencing various oncogenic processes through its interaction with this pathway. This network underscores the intricate molecular mechanisms underpinning the functions of SIRT7, particularly its involvement in regulating key cancer-related signaling pathways, such as PTK2 and NFκB. The insights gained from these analyses may pave the way for developing multitarget therapeutic strategies that could better address the complexity of PCa. Interestingly, efforts to establish reliable biomarkers for PCa diagnosis and progression have prompted researchers to investigate serum-based markers, including SIRT7, with a comparative study evaluating the serum levels of SIRT7 alongside other biomarkers, such as Pentraxin-3 and Fetuin-A, in patients with PCa and benign prostatic hyperplasia [172]. Although SIRT7 levels did not exhibit significant differences between the two groups, the study highlighted its potential utility in conjunction with other biochemical markers. These findings underscore the ongoing need for larger-scale studies to further elucidate the diagnostic value of SIRT7 and its integration into a comprehensive biomarker panel for PCa.

The role of SIRT7 in regulating androgen-induced cellular processes, particularly its effects on autophagy and tumor growth, warrants attention, as studies using LNCaP and 22Rv1 cell lines have shown that SIRT7 depletion reduces cell proliferation and invasion, while increasing sensitivity to radiation therapy [173]. Mechanistically, SIRT7 appears to modulate the AR signaling pathway, where its knockdown upregulates transcription factor SMAD4, further implicating SIRT7 in the regulation of critical oncogenic pathways. The complex interplay between SIRT7 and AR not only positions SIRT7 as a potential prognostic marker but also suggests that targeting SIRT7 could enhance therapeutic responses in PCa, particularly in cases resistant to conventional therapies. Figure 6 summarizes the diverse roles of SIRT7 in influencing the various aspects of prostate carcinogenesis. By elucidating the multifaceted roles of SIRT7, targeted therapies can be developed that effectively tackle the specific challenges associated with aggressive PCa, thereby enhancing patient outcomes in this widespread malignancy.

### 3.2. Tumor Suppressor Sirtuins

#### 3.2.1. SIRT3

SIRT3, a mitochondrial sirtuin, exerts a multifaceted role in the regulation of PCa progression by modulating key metabolic and oncogenic processes. One of its primary functions is to inhibit the acetylation of mitochondrial aconitase (ACO2), a key enzyme involved in mitochondrial metabolism and de novo lipogenesis [174]. In PCa cells, the suppression of SIRT3 by the AR and its coregulator SRC-2 enhances ACO2 activity, promoting citrate synthesis and favoring aggressive cancer phenotypes. Interestingly, acetylation at lysine 258 on ACO2 modulates its function, and SIRT3 reverses this acetylation to suppress tumor progression. SRC-2 depletion, which elevates SIRT3 levels, markedly reduces metastasis, particularly in bone, underscoring the therapeutic potential of targeting this AR-SRC-2-SIRT3 axis in PCa treatment. Beyond its role in ACO2 regulation, SIRT3 also interacts with the steroidogenic enzyme 17β-Hydroxysteroid dehydrogenase type 4 (HSD17B4) [175]. Although HSD17B4 lacks a catalytic function in androgen metabolism, its overexpression in PCa tissues enhances cell proliferation, migration, and invasion. Mechanistically, SIRT3 directly interacts with HSD17B4, inhibiting its acetylation at lysine 669 (K669), a modification that promotes HSD17B4 degradation via chaperone-mediated autophagy. By preventing this acetylation, SIRT3 stabilizes HSD17B4, supporting its oncogenic activity. The acetylation of HSD17B4 is regulated by CREBBP, and DHT treatment exacerbates its acetylation and degradation.

Furthermore, SIRT3 plays a pivotal role in the inhibition of EMT in PCa. Mechanistic studies revealed that SIRT3 suppresses the Wnt/β-catenin signaling pathway, thereby promoting FOXO3A expression, which in turn, inhibits EMT and the migration of PCa cells [93]. Decreased SIRT3 expression correlates with higher Gleason scores and metastatic potential, suggesting that SIRT3 acts as a tumor suppressor by restraining cell migration and invasion. Modulating SIRT3 expression or targeting the Wnt/β-catenin/FOXO3A pathway may provide new avenues for therapeutic interventions in advanced PCa. SIRT3 also modulates key oncogenic signaling pathways, including the PI3K/Akt pathway, to suppress PCa growth. The overexpression of SIRT3 inhibits Akt phosphorylation, leading to the degradation of the oncoprotein c-MYC via ubiquitination, which is crucial for tumor suppression [94]. In contrast, SIRT3 knockdown enhances the growth of PCa cells by maintaining high Akt activity. Moreover, SIRT3’s involvement in necroptosis and the innate immune response further underscores its complex role in PCa progression [166]. SIRT3 also plays a crucial role in the metabolic crosstalk between CAFs and PCa cells, where CAFs undergo Warburg metabolism under the influence of PCa cells, producing lactate that is taken up by cancer cells via monocarboxylate transporter-1 (MCT1). This metabolic symbiosis, driven by hypoxia-inducible factor 1 (HIF1), is regulated by SIRT3, which influences the metabolic reprogramming of both CAFs and PCa cells, promoting cancer cell growth in glucose-deprived environments [176]. The pharmacological inhibition of MCT1 disrupts this symbiotic relationship, pointing to another therapeutic target where SIRT3’s role in metabolic regulation could be exploited.

Collectively, SIRT3 act as tumor suppressor in PCa by regulating key metabolic and oncogenic pathways, including the inhibition of ACO2 acetylation and suppression of the Wnt/β-catenin signaling pathway. Its interactions with the AR and steroidogenic enzyme HSD17B4 enhance aggressive cancer phenotypes and cell survival. Targeting SIRT3 may provide promising therapeutic strategies by restoring its tumor-suppressive functions and improving treatment responses. The collective functions of SIRT3 in PCa are summarized in Figure 7.

#### 3.2.2. SIRT4

SIRT4 has garnered attention for its involvement in tumorigenesis; although, its specific role in PCa remains underexplored. Recent investigations reveal that SIRT4 expression is significantly reduced in PCa tissues compared to non-cancerous tissues, with lower levels correlating with more aggressive tumor phenotypes [177]. Functional assays in the study show that SIRT4 inhibits migration, invasion, and proliferation, while promoting apoptosis, with this suppression mechanistically linked to the inhibition of glutamine metabolism, a critical pathway for sustaining tumor cell growth and invasiveness. SIRT4’s tumor-suppressive role in PCa is further supported by its capacity to modulate post-translational modifications, particularly through ADP-ribosylation. In PCa models, SIRT4’s inhibition of glutamate dehydrogenase 1 (GDH1), a key enzyme in glutamine metabolism, was observed to curb metabolic pathways essential for tumor proliferation [178]. The study reveals that SIRT4 appears to regulate cell cycle progression through its effect on the AKT-p21 axis. By impeding AKT phosphorylation, SIRT4 promotes the nuclear retention of the cell cycle inhibitor p21, thereby enforcing cell cycle arrest and stifling tumor cell division. These findings suggest that SIRT4 not only mediates metabolic rewiring in PCa cells but also exerts direct influence over cell cycle control, marking it as a multifaceted regulator of tumor progression.

Intriguingly, SIRT4’s regulatory activity within the mitochondria extends to its interaction with P21-activated kinase 6 (PAK6) and adenine nucleotide translocase 2 (ANT2), creating a complex interplay between these proteins that govern apoptosis in PCa cells [179]. PAK6, primarily located in the mitochondrial inner membrane, modulates SIRT4 stability through ubiquitin-mediated proteolysis, effectively decreasing its tumor-suppressive activity. In turn, SIRT4 deacetylates ANT2 at K105, promoting its degradation via ubiquitination. This dynamic between the phosphorylation and deacetylation of ANT2 highlights a finely tuned regulatory axis controlled by SIRT4 and PAK6, which together modulate apoptotic pathways critical for maintaining mitochondrial integrity. The PAK6-SIRT4-ANT2 complex, therefore, not only regulates metabolic pathways but also directly influences cell survival and apoptosis in PCa. The mutual regulation of SIRT4 and PAK6 underscores a delicate balance in the modulation of mitochondrial function and apoptotic control in PCa. As PAK6 enhances ANT2 phosphorylation to inhibit apoptosis, it simultaneously destabilizes SIRT4, further tipping the balance toward tumor survival. Clinically, this interplay is reflected in the inverse correlation between PAK6 and SIRT4 expression in PCa tissues, positioning SIRT4 as a critical brake in a system otherwise inclined toward malignancy.

Collectively, these findings position SIRT4 as a key mitochondrial mediator with potential therapeutic relevance, offering novel insights into its function as a metabolic and apoptotic regulator in PCa. Figure 8 illustrates the tumor suppressor role of SIRT4 in PCa.

### 3.3. Janus-Faced/Dual Acting Sirtuin

#### SIRT5

SIRT5 has gained attention in cancer biology due to its involvement in multiple metabolic processes, yet its role in PCa remains insufficiently elucidated. A recent study shows that significantly decreased levels of SIRT5 were identified in more aggressive stages of PCa, with a corresponding reduction in patient survival [95]. The study also identifies a significant increase in succinylation at lysine 118 (K118su) of lactate dehydrogenase A (LDHA), a substrate of SIRT5, which enhances LDH activity and significantly increases the migration, invasion, and LDH activity in PCa cells and patients. The involvement of SIRT5 in PCa progression is further linked to its regulation of the MAPK pathway. Immunohistochemical analyses have revealed that patients with higher Gleason scores exhibited significantly lower SIRT5 expression, and functional assays demonstrated that SIRT5 regulates the activity of the MAPK pathway through the deacylation of acetyl-CoA acetyltransferase 1 (ACAT1), a key metabolic enzyme [96]. By promoting the desuccinylation of ACAT1, SIRT5 effectively suppresses the downstream activation of MAPK-related proteins, such as matrix metalloproteinase 9 (MMP9) and cyclin D1, which are critical for the metastatic potential of PCa cells. Further analysis of SIRT5’s role in PCa metastasis reveals its impact on the PI3K/AKT/NF-ĸB signaling pathway, with proteomic studies using SIRT5 knockout models in PC-3 cells showing a significant increase in pro-inflammatory cytokines, like IL-1β, and the upregulation of the PI3K/AKT/NF-ĸB axis [97]. This elevation in pro-inflammatory cytokines contributes to enhanced migration, invasion, and tumor cell survival, thereby promoting secondary metastasis to distant organs beyond the bone. Acting as both a metabolic regulator and a suppressor of tumor growth and spread, SIRT5 plays a crucial role in modulating key oncogenic signaling pathways, making it a vital player in the metabolic and molecular landscape of PCa (as summarized in Figure 9).

## 4. Sirtuin-Based Interventions in Prostate Cancer Management

Targeting sirtuins presents an exciting strategy in PCa treatment, because of their dynamic expression patterns and their multifaceted roles in regulating cellular processes that are often disrupted in malignancy. Sirtuins have emerged as promising prognostic biomarkers, with their diverse involvement in tumorigenesis and clear correlations to patient outcomes. For instance, SIRT1 is highly expressed in PINs and poorly differentiated carcinomas, positioning it as a strong candidate for prognostic evaluation. Likewise, SIRT6 and SIRT7 have been shown to promote tumor progression, with elevated expression linked to aggressive disease and poor survival, solidifying their potential as reliable biomarkers. SIRT7, in particular, stands out for its association with aggressive tumor phenotypes, offering significant promise in predicting disease prognosis. While the roles of SIRT2, SIRT3, SIRT4, and SIRT5 in PCa progression are becoming clearer, their utility as biomarkers still requires further exploration. Table 2 outlines the specific roles of sirtuins in PCa tumorigenesis and their potential as biomarkers. In the upcoming section, we will delve into the strong link between altered sirtuin expression and PCa aggressiveness, along with therapeutic resistance, which suggests that modulating their activity, particularly SIRT1, SIRT2, SIRT6, and SIRT7, could play a pivotal role in improving treatment outcomes, enhancing the effectiveness of conventional therapies, and addressing the challenges posed by advanced stages of the disease.

### 4.1. SIRT1 Regulators Evaluated Prostate Cancer

Targeting SIRT1 in PCa presents a particularly compelling avenue for therapeutic intervention, owing to its intricate role in modulating critical signaling cascades that influence tumorigenesis. One advantage with SIRT1 is that its deacetylase core lacks intrinsic catalytic activity on its own; instead, the 25 amino acid (ESA) region in the C-terminal domain of SIRT1 serves as an “on switch” for its deacetylase activity. This is significant, because the inhibition of the ESA region, whether by the endogenous inhibitor DBC1 or through the application of mutant peptides, has been shown to enhance chemosensitivity in androgen-refractory PCa cells [181]. Accumulating evidence evaluates SIRT1 suppression as an efficient mechanism for prostate cancer cell. One such study has reported the synthesis of a novel library of indole–triazole derivatives, with one compound, IT-14, demonstrating the significant inhibition of SIRT1’s deacetylation activity, thereby enhancing its potential as druggable target [182]. In vivo analyses revealed that IT-14 effectively mitigated prostatic hyperplasia, as evidenced by a favorable alteration in the prostate weight-to-body weight ratio and preservation of tissue histoarchitecture. Furthermore, the pharmacological inhibition of SIRT1 activity by nicotinamide and splitomycin protected prostate cells from the oncogenic activation of Csn5/Skp2 signaling axis [183]. The classical SIRT1 inhibitor Tenovin-1, combined with the Plk1 inhibitor BI2536, enhances the anticancer efficacy of metformin in a p53-dependent manner [184]. This combination restores glycolytic activity suppressed by metformin, while amplifying its inhibition of oxidative phosphorylation, targeting dysregulated energy metabolism in cancer. Another report suggests that the compound Lead 17 has demonstrated promising efficacy, exhibiting an IC50 of 4.34 μM in suppressing the proliferation of LnCAP cells, alongside significant reductions in reactive oxygen species and pro-inflammatory cytokines [185]. Interestingly, melatonin exerts antitumor effects in PCa by targeting SIRT1-mediated pathways, particularly through transdermal delivery methods, like cryopass-laser treatment. This approach impairs tumor growth by modulating redox balance and influencing SIRT1-regulated proteins, such as PGC-1α, PPARγ, and NFκB, altering the tumor microenvironment [186]. In vivo studies show that oral melatonin reduces tumorigenesis in PCa models, highlighting its potential as a therapeutic agent through SIRT1 modulation [187]. In CRPC, melatonin inhibits tumor progression by restoring CES1 expression, reducing lipid droplet accumulation and reversing ADT resistance through epigenetic modifications [188]. Another interesting study highlights the selective inhibitor 12n as a promising candidate, demonstrating potent inhibition of SIRT1 with an IC50 of 460 nM and remarkable selectivity over other sirtuins [189]. The inhibition of SIRT1 via sodium butyrate induces cellular senescence in PCa cells, manifesting through elevated markers, such as SA-β-Gal and SAHF [190]. This effect is coupled with the downregulation of proto-oncogenes, such as c-Myc and Cyclin D1, alongside a robust upregulation of p21, while p16 expression remains unchanged; additionally, the sodium butyrate-mediated increase in ROS underscores the epigenetic interplay between HDAC inhibition and senescence, highlighting a potent avenue for tumor suppression in PCa.

Resveratrol, a well-known SIRT1 modulator, has been shown to interact with DDX5, promoting its degradation in PCa cells and inducing apoptosis by downregulating the mTORC1 signaling pathway [191]. Furthermore, radiotherapy-induced oxidative damage in PCa cells, marked by reduced SIRT1 levels, nitric oxide, and SOD activity, is counteracted by resveratrol, which attenuates oxidative stress, inhibits caspase-3, and enhances nitric oxide synthase expression to support cell survival [192]. Interesting, the downregulation of SIRT1 in response to radiation is reversed through the pharmacological inhibition of autophagy, elucidating a complex interplay between these pathways [193]. Moreover, the overexpression of SIRT1 markedly alleviates radiation-induced apoptosis, emphasizing its radioprotective effect and underscoring the potential of autophagy-mediated SIRT1 regulation as a promising therapeutic target to enhance the efficacy of PCa treatment. Concurrently, radiation-induced CD105/BMP signaling elevates SIRT1 levels, while the inhibition of SIRT1 expression with TRC105 triggers p53-mediated cell death [194]. Interestingly, targeting SIRT1 can potentially overcome drug resistance in PCa by addressing metabolic adaptations and epigenetic changes associated with chemoresistance. In FLU-resistant PCa cells, increased SIRT1 expression contributes to cancer stem cell-like properties and drug resistance [195]. Additionally, SIRT1 inhibition has been shown to reverse taxane resistance by modulating ABCB1 expression and enhancing cell sensitivity to taxanes [196]. These findings suggest that SIRT1 modulation, combined with epigenetic targeting, could effectively overcome chemoresistance and improve therapeutic outcomes in advanced PCa.

In addition to pharmacological approaches, natural compounds, such as ellagic acid, while inducing oxidative stress and enhancing pro-apoptotic signals, offer a dual mechanism to combat PCa tumor progression [197]. This natural compound, alongside others that target SIRT1, may serve as effective adjunct therapies, potentially leading to improved patient responses and survival rates. Furthermore, the exploration of dietary phytochemicals, such as lovastatin in combination with Antrodia camphorata extract, has revealed synergistic effects in androgen-refractory PCa cells through the inhibition of SIRT1 [198]. One interesting study revealed that Astragalus polysaccharides (APS) significantly reduces SIRT1 expression, impairing lipid metabolism by modulating the AMPK/SREBP1 axis, thereby curbing prostate tumor growth and invasion [199]. Lastly, saffron extract has been shown to downregulate SIRT1 deacetylase activity, contributing to the apoptosis of PCa cells by interfering with DNA repair mechanisms [200]. These innovative approaches highlight the potential of leveraging natural products to inhibit SIRT1 and associated pathways, paving the way for novel therapeutic strategies aimed at mitigating the challenges posed by advanced PCa. Consequently, this finding, along with other targeted approaches, outlines a potential therapeutic strategy where targeting SIRT1, combined with lifestyle interventions like exercise, may help mitigate age-related preneoplastic changes in the prostatic microenvironment.

### 4.2. SIRT2 Regulators Evaluated Prostate Cancer

Targeting SIRT2 has emerged as a promising therapeutic strategy for PCa, particularly for metastatic and treatment-resistant forms of the disease. One pivotal mechanism involves innovative SIRT2 inhibitors, such as sirtuin-rearranging ligands (SirReals), have shown significant therapeutic promise by suppressing both deacetylation and defatty-acylation activities of SIRT2 [201]. These inhibitors have been refined through strategies targeting the prostate-specific membrane antigen (PSMA), enhancing their selective binding and efficacy in PCa cells. Notably, compounds like the PSMA-targeted inhibitor 17 exhibit superior antiproliferative effects and heightened specificity for PCa, overcoming previous limitations in targeting precision. The development of NanoBRET-based assays has further facilitated the quantification of SIRT2 inhibition within cellular contexts, reinforcing the correlation between target engagement and the anticancer efficacy of these inhibitors [202]. Moreover, the pharmacological inhibition of SIRT2, utilizing small molecules, such as oxadiazole-based analogues, has demonstrated efficacy in reducing PCa cell viability and migration, likely by disrupting the deacetylation activity of SIRT2. Docking studies have further elucidated that these inhibitors engage in substrate-competitive and cofactor-noncompetitive interactions with SIRT2, confirming their potential for therapeutic development [203]. Overall, the pharmacological inhibition of SIRT2 not only disrupts tumor growth but also impedes metastatic progression. With their multifaceted actions, SIRT2 inhibitors represent valuable candidates for combating the aggressive and treatment-resistant progression of PCa.

### 4.3. SIRT6 Regulators Evaluated Prostate Cancer

The therapeutic targeting of SIRT6 has emerged as a compelling new avenue for combating metastatic CRPC. Immunohistochemical analyses of PCa tissue microarrays reveal a positive correlation between SIRT6 expression and cancer progression, and subsequent studies demonstrate that silencing SIRT6 through engineered exosomes loaded with small interfering RNA significantly impedes both the proliferation and metastasis of PCa cell lines, both in vitro and in vivo [180]. In parallel, another study has identified quinazolinedione compounds as potent SIRT6 inhibitors, marking a significant leap in therapeutic strategies [204]. These compounds increase histone H3 acetylation at lysine 9, reduce TNF-α production, and enhance glucose uptake in cultured cells. These findings underscore the significant potential of SIRT6-targeting approaches, both as a standalone strategy to suppress tumor progression and as a potent adjunct to existing therapies. While the promise of SIRT6 inhibition marks an exciting frontier in cancer treatment, further research is essential to fully harness its therapeutic potential in prostate cancer.

### 4.4. SIRT7 Regulators Evaluated Prostate Cancer

The intricate relationship between SIRT7 and various molecular pathways underscores its potential as both a therapeutic target and a biomarker in PCa. The emerging evidence suggests that SIRT7 not only promotes tumorigenesis and metastasis but also participates in the modulation of essential cellular processes, such as autophagy and the DNA damage response [205]. Recent studies have explored the efficacy of combinations of existing chemotherapeutics, such as norcantharidin (NCTD) and paclitaxel (PTX), in modulating SIRT7 expression and enhancing anticancer effects [206]. Findings indicate that the NCTD-PTX combination effectively reduces cell viability and induces apoptosis in PCa cells by downregulating SIRT7. This suggests that targeting SIRT7 could synergistically enhance the effectiveness of established chemotherapeutic regimens, providing a viable strategy for addressing the challenges posed by treatment-resistant PCa.

In summary, targeting sirtuins, particularly SIRT1, SIRT2, SIRT6, and SIRT7, presents a multifaceted and promising approach for advancing therapeutic strategies against PCa. By integrating sirtuin modulation into therapeutic regimens, there is a significant opportunity to overcome drug resistance and enhance patient outcomes in advanced PCa, particularly in cases characterized by aggressive disease phenotypes. Table 3 summarizes the specific inhibitors of sirtuins in PCa.

## 5. Conclusions and Perspectives

In conclusion, the sirtuin family exerts a profound influence on PCa by regulating key cellular processes, including metabolic adaptation, epigenetic modulation, oxidative stress response, and DNA repair. Each sirtuin displays distinct functional roles, with SIRT1, SIRT2, SIRT6, and SIRT7, supporting cancer progression by promoting cellular homeostasis, chemoresistance, and AR modulation, particularly in CRPC. In contrast, SIRT3 and SIRT4 act as tumor suppressors by regulating mitochondrial function, inhibiting oncogenic signaling, and inducing apoptosis, thereby suppressing tumor proliferation and invasion. SIRT5 stands out for its dual role, functioning as both a promoter and suppressor in PCa, reflecting the multifaceted nature of sirtuins as tumor-modulating entities. In addition to their role as classical metabolic regulators and histone modifiers, sirtuins also interact with various other epigenetic regulatory mechanisms, including DNA methylation and microRNAs. Notably, only SIRT1 is known to possess such functions. SIRT1 is regulated by miR-34a, leading to drug resistance in hormone-refractory PCa. SIRT1 also interacts with miR-221/222, which is also overexpressed in androgen-independent PCa. The downregulation of miR-212 increases SIRT1 activity, facilitating autophagy and angiogenesis in PCa. SIRT1 also interacts with miR-6068, promoting aggressive phenotypes by inhibiting the HIC2/SIRT1 axis in PCa cells. SIRT1 also plays a role in regulating DNA methylation processes in PCa, with increased SIRT1 expression in the TRAMP mouse model, correlating with reduced levels of the methylation regulator *HIC1*. Additionally, SIRT1 interacts with the epigenetic regulator MPP8 to promote histone deacetylation and gene silencing through H3K9 methylation, which facilitates EMT. Recent studies also emphasize the role of SIRT1 in modulating DNA methylation through interactions with NNMT and the PRC4 complex. However, these additional mechanisms of epigenetic regulation are currently only limited to SIRT1, highlighting the need for further research to unravel the complex interactions between other sirtuins and epigenetic mechanisms in PCa.

Another important aspect of PCa that sirtuins contribute to is the modulation of resistance mechanisms in CRPC, primarily through epigenetic regulation, DNA repair modulation, and adaptive responses to therapeutic stress. SIRT1 interaction with KU70 facilitates NHEJ-mediated DNA repair, supporting resistance mutation acquisition, while its regulation of autophagy aids in adapting to radiation-induced DNA damage. SIRT2 contributes to resistance by modulating histone acetylation (e.g., H3K18), and SIRT7 drives resistance and metastasis through epigenetic reprogramming associated with EMT. Targeting these sirtuins and their regulatory pathways offers a promising approach to overcoming resistance and improving therapeutic outcomes. Moreover, future research should focus on unraveling the complex interactions between other sirtuins, particularly SIRT5 and SIRT6, to validate their potential role in modulating the PCa tumor microenvironment in the context of drug treatment.

As PCa therapy continues to advance, integrating sirtuin modulation into standard care could revolutionize treatment outcomes, particularly through combination therapies targeting both cancer metabolism and immune responses, especially in advanced or drug-resistant PCa patients. Also, investigating natural compounds and synthetic agents that selectively target sirtuin pathways could create novel adjunct therapies, potentially increasing sensitivity to chemotherapy and overcoming drug resistance. Additionally, there is also an urgent need to advance preclinical and clinical research on sirtuin-based inhibitors to better understand their pleiotropic effects, optimize their specificity, and assess their safety and efficacy in cancer treatment. These efforts are essential for translating promising preclinical results into effective therapeutic strategies for resistant cancers, like prostate cancer. Moreover, at the mechanistic level, while the roles of sirtuins, such as SIRT1, SIRT2, and SIRT7, have been well explored in PCa, there is a critical need to investigate other sirtuins, like SIRT-3, SIRT4, SIRT5, and SIRT6. A deeper understanding of sirtuin functions could open new therapeutic avenues for targeting prostate cancer. Another promising area for future research is exploring the interaction of sirtuins with miRNAs as upstream regulators, co-regulators, or downstream targets. Sirtuin-based therapies could be combined with other treatments, such as ADT or immunotherapy. However, to fully realize their potential, it is essential to address the limitations of these approaches, particularly their pleiotropic effects, which may play diverse and sometimes opposing roles in cellular pathways. Achieving tissue-specific targeting remains challenging, as sirtuins are ubiquitously expressed and regulate both tumor-promoting and tumor-suppressive processes. Furthermore, the risk of adaptive resistance to sirtuin modulators highlights the need for combination strategies and biomarker-guided therapies. Ultimately, a comprehensive understanding of sirtuin-mediated pathways in prostate cancer’s signaling landscape may lead to the development of personalized, sirtuin-targeting therapies, improving prognosis and quality of life for patients with aggressive prostate cancer.

## Figures and Tables

**Figure 1 ijms-26-00183-f001:**
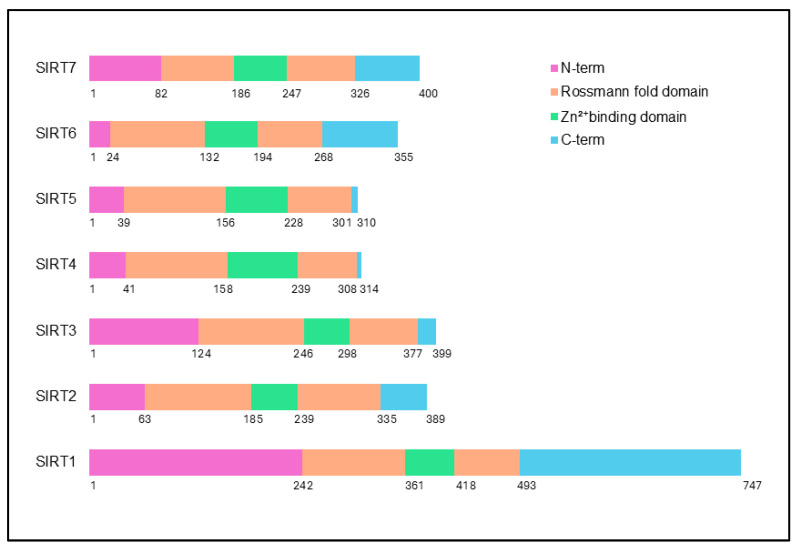
Structural features of sirtuins. All sirtuins share a conserved catalytic core, comprising a Rossmann fold domain, a Zn^2^^+^-binding domain, and a catalytic histidine, which are critical for their enzymatic functions. Despite this shared core, human sirtuins feature unique N-terminal and C-terminal domains, which vary in length and sequence, contributing to their diverse roles.

**Figure 2 ijms-26-00183-f002:**
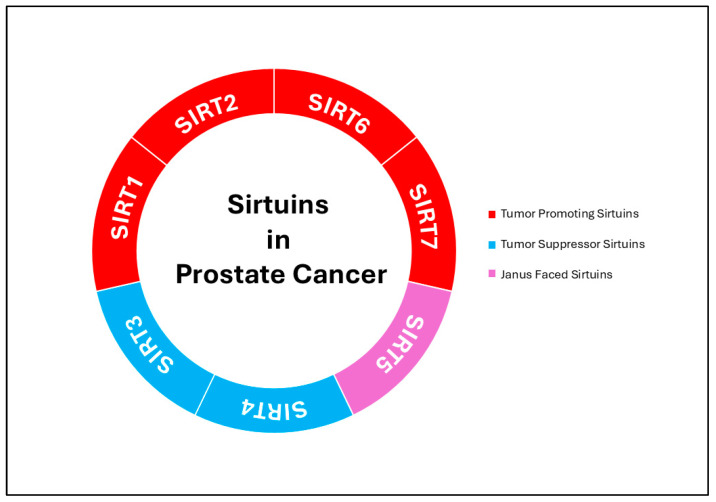
Categorization of sirtuins in prostate cancer based on their tumor-modulating roles. Sirtuins that promote prostate tumor progression include SIRT1, SIRT2, SIRT6, and SIRT7 (highlighted in red). Those with tumor-suppressive functions are SIRT3 and SIRT4 (highlighted in blue). SIRT5 exhibits a Janus-faced role, acting as both a tumor suppressor and promoter (highlighted in purple).

**Figure 3 ijms-26-00183-f003:**
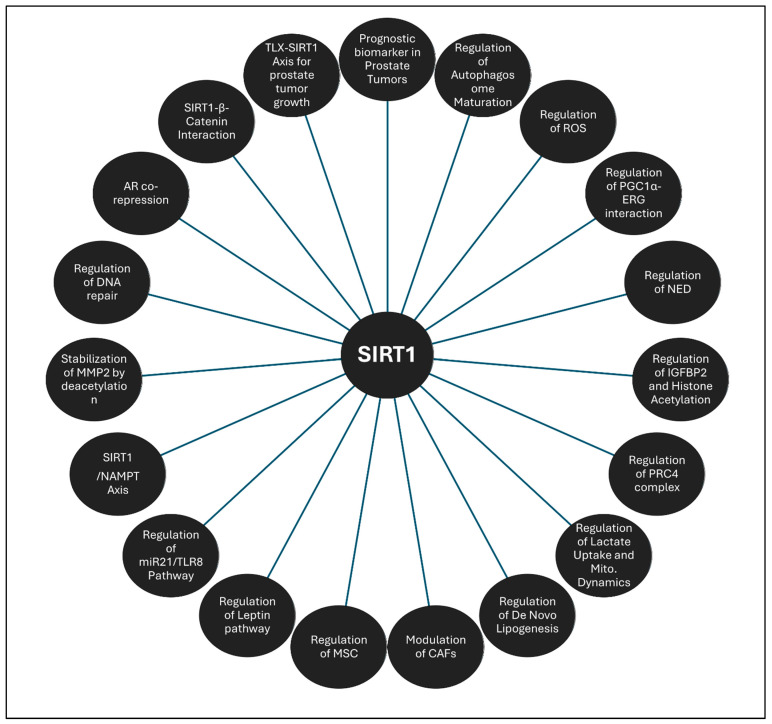
Multifaceted role of SIRT1 in PCa. SIRT1 maintains cellular homeostasis, mitigating PIN development by facilitating autophagosome maturation. SIRT1 preserves mitochondrial integrity by reducing ROS levels through the regulation of SOD2 acetylation and promotes antioxidant defense via the ERG-PGC1α pathway. Additionally, SIRT1 is involved in NED by activating the Akt and AMPK-SIRT1 pathways, particularly in response to oxidative stress and inflammatory signaling from ADT. Epigenetically, SIRT1 modulates gene expression by controlling IGFBP2 through histone acetylation and contributes to chromatin remodeling via the PRC4 complex, promoting oncogenic transformation. In terms of metabolism, SIRT1 drives mitochondrial biogenesis and de novo lipogenesis, supporting lipid synthesis critical for cancer cell growth. It also modulates immune responses, facilitating immune evasion, while recruiting NK cells and macrophages in MSCs to counter tumor proliferation. Under hypoxia, SIRT1 enhances cellular adhesion and invasiveness through the leptin-HIF-1α pathway, crucial for tumor spread. Moreover, SIRT1 regulates apoptosis by deacetylating FOXO transcription factors, stabilizing MMP2 for EMT and supporting therapeutic resistance through KU70 interactions. Its role as an AR corepressor aids in resistance to ADT, particularly in castration-resistant cases. Lastly, SIRT1’s interaction with TLX and β-catenin impacts cell survival and metabolism, highlighting its complex regulatory impact in PCa progression and treatment resistance.

**Figure 4 ijms-26-00183-f004:**
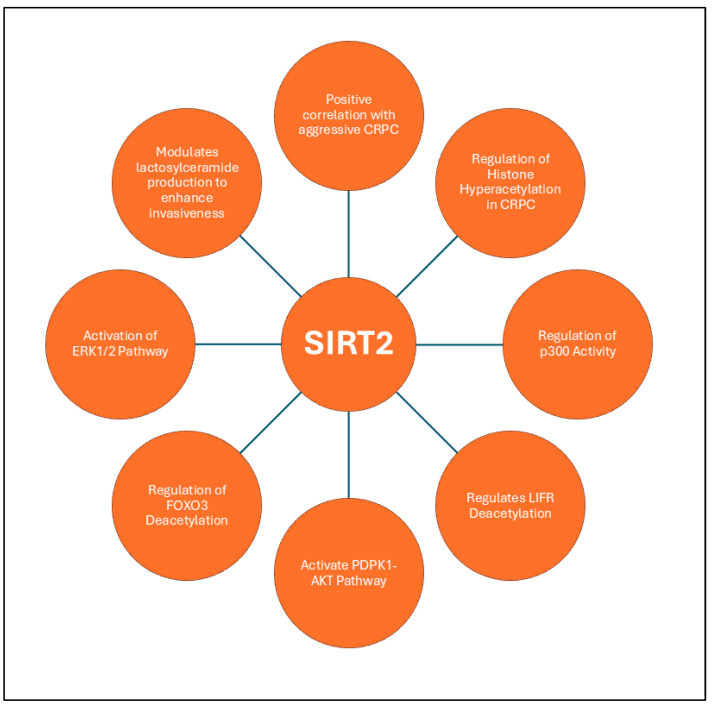
Role of SIRT2 in prostate cancer progression. In PCa, SIRT2 exhibits a complex role, with its expression levels and function shifting from tumor-suppressive in early stages to potentially oncogenic in advanced stages, like CRPC and NEPC. Initially, SIRT2 helps maintain epigenetic stability by deacetylating histones, particularly countering the hyperacetylation of H3K18 seen in aggressive tumors. This reduction in SIRT2 activity in CRPC correlates with increased acetylation by p300, contributing to oncogenic gene expression and AR signaling resistance. SIRT2 also influences key signaling molecules, such as by deacetylating the LIFR to suppress oncogenic signaling through the PDPK1-AKT pathway. Furthermore, SIRT2 modulates transcription factors, like FOXO3, accelerating its degradation and, thus, reducing cell cycle arrest and apoptosis, particularly in CRPC and NEPC. Additionally, SIRT2 supports metabolic adaptations by promoting the production of lactosylceramide, which enhances cancer cell invasiveness.

**Figure 5 ijms-26-00183-f005:**
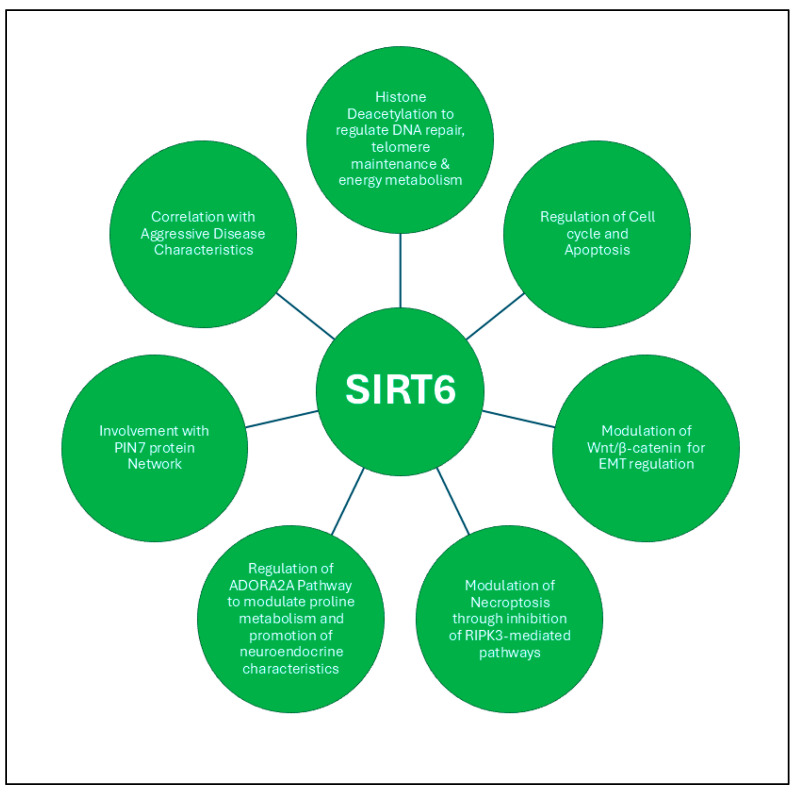
Diverse roles of SIRT6 in prostate cancer. SIRT6 is frequently overexpressed in PCa, correlating with aggressive disease traits, such as high Gleason scores and metastasis. Knockdown studies show that silencing SIRT6 reduces cell viability, induces apoptosis, and increases DNA damage, underscoring its role in tumor progression. SIRT6 promotes cancer cell proliferation, migration, and invasion, potentially by activating the Wnt/β-catenin pathway, a driver of EMT and metastasis. Furthermore, SIRT6’s modulation of necroptosis impacts immune cell recruitment within the tumor microenvironment, enhancing inflammatory responses upon SIRT6 inhibition. SIRT6 also contributes to lineage plasticity in neuroendocrine differentiation, particularly through ADORA2A-driven metabolic rewiring and regulates glycolytic activity via E2F1, which suppresses SIRT6 expression.

**Figure 6 ijms-26-00183-f006:**
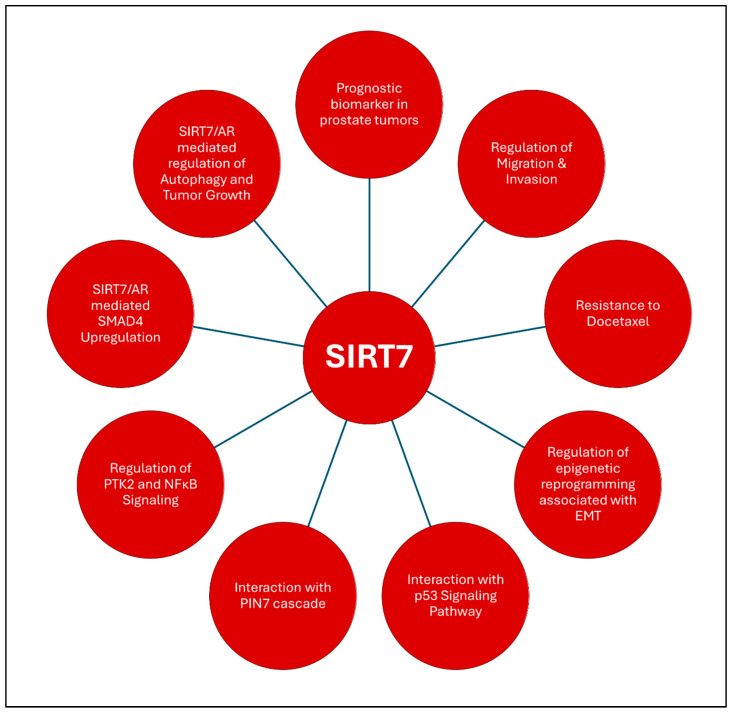
SIRT7 as modulator of prostate cancer progression. Elevated expression of SIRT7 in tumor tissues correlates positively with cancer grade and is linked to increased migration and invasion in androgen-independent PCa cell lines, while its silencing reduces these aggressive traits. Notably, SIRT7 overexpression in less aggressive cell lines enhances resistance to the chemotherapeutic agent docetaxel, underscoring its role in promoting treatment resistance. SIRT7 is also involved in epigenetic reprogramming associated with EMT, contributing to metastatic potential and poor patient prognosis. The interconnection of SIRT7 with critical signaling pathways, particularly the p53 pathway, highlights its influence on various oncogenic processes. Additionally, SIRT7’s regulation of AR signaling suggests its potential as a prognostic marker and therapeutic target, especially in treatment-resistant PCa cases.

**Figure 7 ijms-26-00183-f007:**
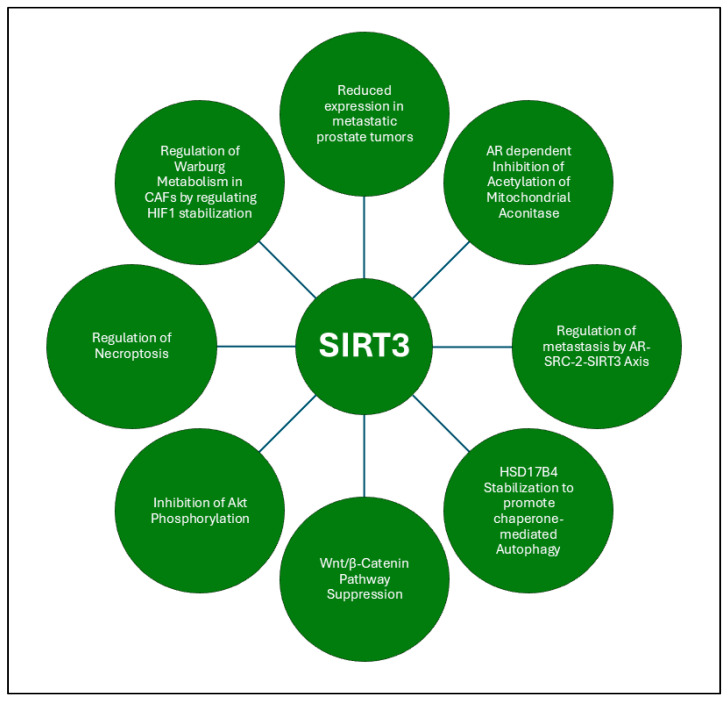
Tumor-suppressive role of SIRT3 in prostate cancer. SIRT3 inhibits the acetylation of mitochondrial ACO2, enhancing its activity and promoting citrate synthesis, which favors aggressive cancer phenotypes. The suppression of SIRT3 by the AR and its co-regulator SRC-2 leads to increased ACO2 activity, while SIRT3 overexpression reduces metastasis, particularly to bone, highlighting the therapeutic potential of targeting this AR-SRC-2-SIRT3 axis. Furthermore, SIRT3 interacts with the steroidogenic enzyme HSD17B4, preventing its acetylation and subsequent degradation, thus supporting its oncogenic activity in PCa. SIRT3 also inhibits EMT by suppressing the Wnt/β-catenin signaling pathway, thereby promoting FOXO3A expression, which correlates with reduced cell migration and invasion. Additionally, SIRT3 inhibits the PI3K/Akt pathway, leading to c-MYC degradation, further establishing its tumor-suppressive role. Moreover, SIRT3’s regulation of necroptosis and its impact on the metabolic interplay between CAFs and PCa cells suggest that it influences cancer cell survival and growth.

**Figure 8 ijms-26-00183-f008:**
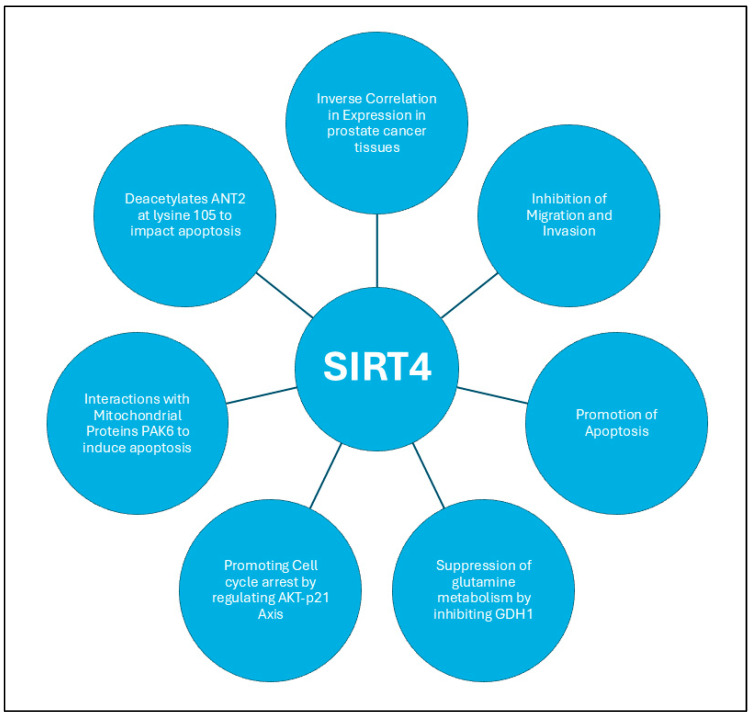
Role of SIRT4 in prostate cancer. SIRT4 expression is significantly reduced in PCa tissues compared to non-cancerous counterparts, with lower levels correlating with more aggressive tumor characteristics, such as higher Gleason scores. Functional assays reveal that SIRT4 inhibits the migration, invasion, and proliferation of PCa cells, while promoting apoptosis, primarily by disrupting glutamine metabolism, which is crucial for tumor growth. Mechanistically, SIRT4 hinders GDH1, limiting metabolic pathways vital for tumor cell proliferation. Additionally, SIRT4 influences cell cycle progression by impeding AKT phosphorylation, thereby enhancing the nuclear retention of the cell cycle inhibitor p21, which leads to cell cycle arrest. The interplay between SIRT4 and PAK6 further complicates its role; while SIRT4 deacetylates ANT2 to promote its degradation and regulate apoptosis, PAK6 destabilizes SIRT4, creating a regulatory feedback loop that favors tumor survival.

**Figure 9 ijms-26-00183-f009:**
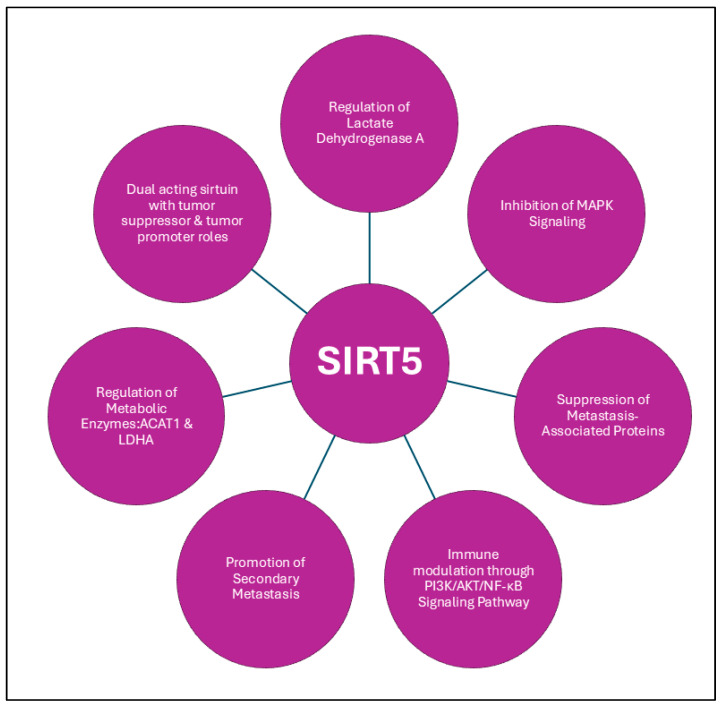
Dual role of SIRT5 in prostate cancer. SIRT5 levels are markedly decreased in aggressive stages of PCa, correlating with reduced patient survival. Its desuccinylation activity is particularly critical, as it targets LDHA, where increased succinylation at lysine 118 enhances LDH activity, promoting the migration and invasion of cancer cells. Additionally, SIRT5 regulates the MAPK pathway by desuccinylating ACAT1, thereby inhibiting downstream targets, like matrix MMP9 and cyclin D1, both vital for metastatic potential. Furthermore, SIRT5 influences the PI3K/AKT/NF-ĸB signaling pathway, where its loss leads to increased pro-inflammatory cytokines and enhanced tumor cell survival, contributing to metastasis beyond the bone.

**Table 1 ijms-26-00183-t001:** Zinc-binding cysteine positions in sirtuins.

Sirtuin	Zinc-Binding Cysteine Positions
SIRT1	Cys370, Cys373, Cys396, Cys399
SIRT2	Cys195, Cys200, Cys221, Cys224
SIRT3	Cys255, Cys258, Cys280, Cys283
SIRT4	Cys159, Cys162, Cys220, Cys223
SIRT5	Cys166, Cys169, Cys207, Cys212
SIRT6	Cys139, Cys142, Cys166, Cys177
SIRT7	Cys194, Cys197, Cys225, Cys228

**Table 2 ijms-26-00183-t002:** Specific role of sirtuins in prostate cancer tumorigenesis and their potential as biomarkers for prostate cancer.

Sirtuin	Significance in Prostate Tumorigenesis	Prospects as a Prognostic Biomarker in Prostate Cancer	References
SIRT1	The expression was markedly elevated in PINs and poorly differentiated carcinomas.	Yes	[99,100]
SIRT2	The expression is elevated in carcinomas and is associated with tumor progression.	Not yet established.	[91]
SIRT3	Low expression contributes to tumor development, progression, and poor prognosis.	Not yet established.	[93,174,175]
SIRT4	Reduced expression in carcinomas promotes evasion of apoptosis and enhanced cell survival.	Not yet established.	[177]
SIRT5	Exhibits a dual role as both a tumor promoter and suppressor in prostate tumorigenesis.	Not yet established.	[95]
SIRT6	Elevated expression correlates with aggressive disease characteristics, including increased nodal metastasis.	Yes	[162,180]
SIRT7	Elevated expression is linked to aggressive tumor phenotypes and unfavorable patient outcomes.	Yes	[98]

**Table 3 ijms-26-00183-t003:** Summary of targetable sirtuins and their modulators in prostate cancer.

Sirtuin	Inhibitor of Sirtuin	Mode of Inhibitor Action	Reference Number
SIRT1	DBC1	Enhances chemosensitivity by inhibiting the ESA region of SIRT1.	[181]
	Splitomycin	Inhibits oncogenic activation of Csn5/Skp2 signaling.	[183]
	IT-14	Inhibits SIRT1’s deacetylation activity, reducing prostatic hyperplasia.	[182]
	Tenovin-1	Combined with Plk1 inhibitor BI2536 to augment anti-neoplastic efficacy of metformin.	[184]
	Lead 17	Suppress proliferation of PCa cells and reduces ROS and pro-inflammatory cytokines.	[185]
	Melatonine	Exerts antitumor effects in PCa by influencing epigenetic modification and redox balance.	
	Selective inhibitor 12n	Competitive inhibition of acetyl peptide substrates and noncompetitive towards NAD^+^.	[189]
	Sodium butyrate	Induces senescence and alters proto-oncogene expression in PCa cells.	[190]
	Resveratrol	Triggers apoptosis by degrading DDX5 and inhibiting mTORC1 signaling.	[191]
	TRC105	Antagonizing CD105 mediated elevation of SIRT1 and induces p53-mediated cell death.	[194]
	Ellagic acid	Downregulates SIRT1 and induces oxidative stress to promote apoptosis.	[197]
	Combinatorial treatment of lovastatin and Antrodia camphorata extract	Enhances cell cycle arrest and apoptosis and reduces stemness in PCa.	[198]
	Astragalus polysaccharides (APS)	Reduces SIRT1 expression and disrupts lipid metabolism.	[199]
	Saffron extract	Downregulates SIRT1, contributing to apoptosis through DNA repair interference.	[200]
	Cancer-derived exosomal microRNA-1275	Downregulates SIRT2 to enhance osteoblast activity, facilitating bone metastasis.	[207]
SIRT2	Sirtuin-rearranging ligands (SirReals)	Suppresses deacetylation and defatty-acylation, reducing c-Myc levels.	[201]
	Oxadiazole-based analogues	Reduces PCa cell viability and migration by disrupting SIRT2 activity.	[203]
SIRT6	Quinazolinedione compounds, E3 aptamer-modified siRNA loaded exosomes based inhibition	Inhibits SIRT6, increasing histone acetylation and enhancing glucose uptake.	[180,204]
SIRT7	NCTD-PTX combination	Downregulates SIRT7, inducing apoptosis and reducing cell viability.	[206]

## Data Availability

Not applicable.

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
