# Peer review of "Molecular Sentinels: Unveiling the Role of Sirtuins in Prostate Cancer Progression"

_ijms, 2024, doi:10.3390/ijms26010183_

Round 1

Reviewer 1 Report

Comments and Suggestions for Authors

This manuscript by Chouhan and Kumar is a review that presents a detailed examination of the dualistic roles of sirtuins in prostate cancer (PCa), highlighting their functions as both tumor suppressors and promoters depending on the cellular context. This comprehensive approach helps us understand the complex nature of sirtuins in cancer biology. Even though this is a highly scholarly work encompassing 217 references and represents a substantial effort in writing, the manuscript can benefit from a significant revision. The authors should consider dividing this review into two manuscripts due to its length (53 pages, including references), verbosity in some areas, repetitive statements and details irrelevant to the objective. For example, section 2 (Structural variations among sirtuins and their functional implications) should be deleted or reduced to 2-3 paragraphs. In some sentences (l.83), the role of SIRT7 related to breast and liver cancers is discussed; it is important to focus on PCa. This manuscript has a wealth of information and strengths, including evaluating seven sirtuins on diverse pathways such as metabolic regulation, chromosomal stability, inflammation and cellular survival. Another strength is examining specific molecular mechanisms by which sirtuins influence the progression of PCa. Some sirtuins like SIRT1 are evaluated as potential biomarkers of lethal PCa. The authors discuss the role of sirtuins, including protein targeting, in drug resistance (e.g. androgen deprivation therapy), which causes castrate-resistant PCa. This is the most lethal and incurable form of the disease. The potential for therapeutic targets is also presented in this review.

In addition to the comments presented in the first paragraph, the review has some limitations that the authors might want to consider before resubmission. While the review discusses the roles of various sirtuins in PCa metabolism and progression, it appears to focus more on SIRT1, SIRT2, and SIRT3, potentially overlooking other sirtuins (e.g.  SIRT4, SIRT5, and SIRT6). However, perhaps this is perhaps a reflection of gaps in our current knowledge. Reducing the length of the manuscript might provide the authors the opportunity for a deeper discussion of how sirtuins contribute to resistance mechanisms in CRPC and the critical role of regulation of DNA repair. The latter is a major gap in this review and a critical mechanistic area associated with the progression of PCa and the development of CRPC. The review would also benefit from a deeper analysis of the therapeutic potential of sirtuins. It is unclear whether sirtuins interact with other epigenetic regulatory mechanisms such as DNA methylation and miRNA expression.

Author Response

Reviewer 1: Comment 1: This manuscript by Chouhan and Kumar is a review that presents a detailed examination of the dualistic roles of sirtuins in prostate cancer (PCa), highlighting their functions as both tumor suppressors and promoters depending on the cellular context. This comprehensive approach helps us understand the complex nature of sirtuins in cancer biology. Even though this is a highly scholarly work encompassing 217 references and represents a substantial effort in writing, the manuscript can benefit from a significant revision. The authors should consider dividing this review into two manuscripts due to its length (53 pages, including references), verbosity in some areas, repetitive statements and details irrelevant to the objective. For example, section 2 (Structural variations among sirtuins and their functional implications) should be deleted or reduced to 2-3 paragraphs. In some sentences (l.83), the role of SIRT7 related to breast and liver cancers is discussed; it is important to focus on PCa. This manuscript has a wealth of information and strengths, including evaluating seven sirtuins on diverse pathways such as metabolic regulation, chromosomal stability, inflammation and cellular survival. Another strength is examining specific molecular mechanisms by which sirtuins influence the progression of PCa. Some sirtuins like SIRT1 are evaluated as potential biomarkers of lethal PCa. The authors discuss the role of sirtuins, including protein targeting, in drug resistance (e.g. androgen deprivation therapy), which causes castrate-resistant PCa. This is the most lethal and incurable form of the disease. The potential for therapeutic targets is also presented in this review.

Author’s reply to comment 1- Thank you for your thoughtful and constructive comments. We greatly appreciate the time and effort you invested in providing detailed feedback, which significantly helps us improve the manuscript. We acknowledge that the length of the manuscript is substantial. However, we aimed to provide a holistic perspective on the dualistic roles of sirtuins in prostate cancer (PCa). While dividing the manuscript into two separate reviews could provide greater focus, we believe that maintaining a unified document offers a more cohesive narrative and comprehensive understanding of sirtuins in PCa. Nevertheless, we have now thoroughly streamlined the content to ensure clarity and conciseness in the revised manuscript. We understand your concern regarding the length of Section 2. We have now condensed this section to highlight the most relevant structural features of sirtuins (Highlighted section: line-103-142). Regarding pointing out the discussion of SIRT7 in breast and liver cancers, we have now carefully revised such sections to ensure that the primary focus remains on PCa (Highlighted section: line-76-89). We appreciate your acknowledgment of the manuscript’s strengths we ensure that these aspects remain well-articulated and emphasized in the revised manuscript. We have thoroughly revised the manuscript to eliminate verbosity and remove repetitive statements, ensuring it aligns with your valuable feedback. These revisions aim to strike a balance between comprehensive coverage and readability. Thank you once again for your constructive critique, which has been instrumental in refining and strengthening our work.

 Comment 2: In addition to the comments presented in the first paragraph, the review has some limitations that the authors might want to consider before resubmission. While the review discusses the roles of various sirtuins in PCa metabolism and progression, it appears to focus more on SIRT1, SIRT2, and SIRT3, potentially overlooking other sirtuins (e.g.  SIRT4, SIRT5, and SIRT6). However, perhaps this is perhaps a reflection of gaps in our current knowledge. Reducing the length of the manuscript might provide the authors the opportunity for a deeper discussion of how sirtuins contribute to resistance mechanisms in CRPC and the critical role of regulation of DNA repair. The latter is a major gap in this review and a critical mechanistic area associated with the progression of PCa and the development of CRPC. The review would also benefit from a deeper analysis of the therapeutic potential of sirtuins. It is unclear whether sirtuins interact with other epigenetic regulatory mechanisms such as DNA methylation and miRNA expression.

Author’s reply to comment 2- Thank you for your thoughtful and constructive feedback. Regarding the focus on SIRT1, SIRT2, and SIRT3, we acknowledge that relatively limited discussion of other sirtuins, such as SIRT4, SIRT5, and SIRT6 is indeed reflective of the current gaps in existing literature on their roles in PCa progression. Moreover, we believe that this will be a great point to discuss in conclusion section. Thus, we have now made an effort to highlight these gaps explicitly and discuss the potential significance of these lesser-studied sirtuins in Conclusion and perspectives section in revised manuscript (Highlighted section: line-1026-1029). We also take your point about the length of the manuscript and have now made concerted efforts to reduce and condense it wherever possible in the revised version. At the same time, we have expanded our analysis of sirtuins’ contributions to resistance mechanisms in CRPC and their role in regulating DNA repair (Highlighted section: lines-260-263, 295-300, 340-345, 418-421, 489-502, 514-517, 602-613, 867-869, 890-899, 909-914). Furthermore, recognizing the importance of this topic, we have elaborated on the significance of sirtuins in resistance mechanisms within the Conclusion and Perspectives section to provide a more comprehensive discussion (Highlighted section: line-1004-1015). Additionally, we have thoroughly revised section 4 and now provided a more detailed to table 3 (Highlighted section: line-814-996) and discussion of the therapeutic potential of sirtuins including potential limitations of SIRT-based therapeutics in conclusion section (Highlighted section: line-1016-1041). Also, we have now included a new Table (Table 2 in the revised manuscript) that summarizes the specific roles of sirtuins in prostate tumorigenesis and identifies potential sirtuins as biomarkers for prostate cancer. (Highlighted section, Table 2, lines-834). We have now also highlighting the interactions of individual sirtuins with other epigenetic regulatory mechanisms such as miRNA expression and DNA methylation within each sirtuins respective subsections (Highlighted section: lines- 280-233, 409-456, 204-209, 320-326, 446-449) and in conclusion section (Highlighted section: lines-986-1003). We hope that the revisions made throughout indicated sections in the manuscript adequately address the reviewer’s concerns. Thank you again for your valuable comments, which have greatly enhance the quality of our review.

Reviewer 2 Report

Comments and Suggestions for Authors

In their paper, the authors addressed prostate cancer and the impact of sirtuins on disease progression.

In the introduction, the authors briefly described prostate cancer and introduced sirtuins. I think the paper would have benefited from an in-depth description of prostate cancer and a more compact description of sirtuins, given that they are described in detail later in the paper.

 In paragraph 2nd, the authors listed the localization of zinc-binding cysteines in individual sirtuins. I think it would be much clearer to present them in a table.

Also, it would be beneficial to put the location of sirtuins in the table.

Authors should improve the structure of the work. For example, I think that a cursory mention of each sirtuin at the beginning of paragraph 3rd is not needed and unnecessarily lengthens an already long work.

Moreover, the paragraph 3.1.1 is very long and it is hard to maintain focus on the text. I think it is necessary to introduce subsections that would organize this part. The same applies to paragraph 4th.

In paragraph 3.1.2, in the first and second paragraphs, the authors write about the same thing. I think they should combine and systemize them.

There are minor editorial errors in the work, e.g. in verse 64, 115 or 673, etc.

Author Response

In their paper, the authors addressed prostate cancer and the impact of sirtuins on disease progression.

Comment 1: In the introduction, the authors briefly described prostate cancer and introduced sirtuins. I think the paper would have benefited from an in-depth description of prostate cancer and a more compact description of sirtuins, given that they are described in detail later in the paper.

Author’s reply to comment 1- Thank you for your valuable feedback. We understand your concern regarding the balance between the descriptions of prostate cancer and sirtuins in the introduction. In the revised manuscript, we have now provided a more in-depth description of prostate cancer, focusing on its molecular and clinical aspects, to better contextualize the relevance of sirtuins in this disease (Highlighted section: line-52-63). Simultaneously, we have condensed the introduction of sirtuins to avoid redundancy (Highlighted section: line-76-89). We believe these new adjustments have enhanced the clarity and focus of the introduction.

Comment 2: In paragraph 2nd, the authors listed the localization of zinc-binding cysteines in individual sirtuins. I think it would be much clearer to present them in a table. Also, it would be beneficial to put the location of sirtuins in the table.

Author’s reply to comment 2- Thank you for your thoughtful suggestion. In the revised manuscript, we have now created new Table 1 and listed the localization of zinc-binding cysteines on each sirtuin, which we believe now provide a more concise and visually accessible format for readers (Highlighted section: line-114). We have significantly condensed this section while preserving the essential information (Highlighted section: line-103-142) and believe these revisions adequately address the reviewers' concerns.

Comment 3: Authors should improve the structure of the work. For example, I think that a cursory mention of each sirtuin at the beginning of paragraph 3rd is not needed and unnecessarily lengthens an already long work.

Author’s reply to comment 3- Thank you for your feedback. In the revised version, we have now significantly reduced concerning paragraph 3 (Highlighted section: line-76-89) We now believe that this adjustment helps to reduce the length of the manuscript and maintains focus on the more detailed discussions presented later in the paper.

Comment 4: Moreover, the paragraph 3.1.1 is very long and it is hard to maintain focus on the text. I think it is necessary to introduce subsections that would organize this part. The same applies to paragraph 4th.

Author’s reply to comment 4- Thank you for pointing this out. In the revised manuscript, we have introduced subsections within paragraph 3.1.1 to better organize the content and improve readability (Highlighted section: line-185-462). Similarly, we have restructured paragraph 4 and section 4 by adding subsections to provide a clearer and more logical flow (Highlighted section: line-328-381). We believe these changes will enhance the overall structure and readability of the manuscript.

Comment 5: In paragraph 3.1.2, in the first and second paragraphs, the authors write about the same thing. I think they should combine and systemize them.

Author’s reply to comment 5- Thank you for your observation. In the revised manuscript, we have combined and streamlined these paragraphs (Highlighted section: line-487-502). We believe this adjustment enhances clarity and improves the overall flow of the section.

Comment 6: There are minor editorial errors in the work, e.g. in verse 64, 115 or 673, etc.

Author’s reply to comment 6- We have carefully reviewed the manuscript and corrected all identified editorial errors (Highlighted section: line-84, 109, 551). We have now made rigorous efforts to ensure consistency and accuracy throughout the text. We appreciate your attention to details and hope all the above-mentioned updates in the revised manuscript will adequately address the reviewer’s concerns.

Reviewer 3 Report

Comments and Suggestions for Authors

I would like to congratulate the authors on their manuscript, which focuses on the dualistic roles of sirtuins—a family of NAD+-dependent histone deacetylases—in the pathogenesis of PCa, carefully dissecting the unique contributions of each sirtuin to either tumor suppression or progression, depending on the cellular context.

--------------------------------------

More comments following the MDPI guided points

1. How do you consider the topic original or relevant to the field? Does it address a specific gap in the field? Please also explain why this is/is not the case.
The article is highly relevant as it explores the roles of sirtuins in the context of prostate cancer, a topic of increasing interest due to the growing body of research in recent years. Despite the number of published studies on this topic, this review stands out by synthesizing the roles of individual sirtuins with a specific focus on their physiological functions and their contribution to the pathogenesis of prostate cancer. This focused approach addresses a critical gap in the field by consolidating disparate findings into a coherent narrative.
2. What does it add to the subject area compared with other published material?
This review distinguishes itself from other publications by concentrating solely on the pathophysiology of prostate cancer while providing a detailed analysis of the physiological roles of each sirtuin. Its structured and didactic presentation makes complex mechanisms accessible, potentially aiding both researchers and clinicians in understanding the therapeutic implications of targeting sirtuins.

3. If possible, what specific improvements should the authors consider adding regarding each topic discussed in this review?

Expanding the discussion on translational aspects, such as ongoing clinical trials or experimental therapies targeting sirtuins, would strengthen the review’s practical relevance.

4. Are the conclusions consistent with the evidence and arguments presented, and do they address the main question posed? Please also explain why this is/is not the case.
Yes, the conclusions are consistent with the evidence and arguments presented. The review successfully synthesizes the role of sirtuins in prostate cancer pathogenesis and aligns this understanding with the broader therapeutic potential of targeting these molecules. The main questions posed by the authors are adequately addressed through a logical and evidence-based discussion.

5. Are the references appropriate?
The references are appropriate and relevant

Author Response

Comment 1: I would like to congratulate the authors on their manuscript, which focuses on the dualistic roles of sirtuins—a family of NAD+-dependent histone deacetylases—in the pathogenesis of PCa, carefully dissecting the unique contributions of each sirtuin to either tumor suppression or progression, depending on the cellular context.

Author’s reply to comment 1- Thank you very much for your kind words and thoughtful feedback. We are glad that the manuscript effectively dissects the unique contributions of each sirtuin in tumor suppression and progression, and we value your positive assessment of our approach.

Comment 2: More comments following the MDPI guided points

  1. How do you consider the topic original or relevant to the field? Does it address a specific gap in the field? Please also explain why this is/is not the case.

The article is highly relevant as it explores the roles of sirtuins in the context of prostate cancer, a topic of increasing interest due to the growing body of research in recent years. Despite the number of published studies on this topic, this review stands out by synthesizing the roles of individual sirtuins with a specific focus on their physiological functions and their contribution to the pathogenesis of prostate cancer. This focused approach addresses a critical gap in the field by consolidating disparate findings into a coherent narrative.

Author’s reply to comment 2- Thank you for your thoughtful feedback. We believe that our focused approach in this review manuscript to comprehensively discus sirtuins' roles in prostate cancer addresses a critical gap and adds value to the field.

Comment 3: 2. What does it add to the subject area compared with other published material?

This review distinguishes itself from other publications by concentrating solely on the pathophysiology of prostate cancer while providing a detailed analysis of the physiological roles of each sirtuin. Its structured and didactic presentation makes complex mechanisms accessible, potentially aiding both researchers and clinicians in understanding the therapeutic implications of targeting sirtuins.

Author’s reply to comment 3- Thank you for your positive feedback. We’re glad our focused and structured approach in this review manuscript will enhance the understanding of sirtuins' roles in prostate cancer and their therapeutic implications.

Comment 4: 3. If possible, what specific improvements should the authors consider adding regarding each topic discussed in this review?

Expanding the discussion on translational aspects, such as ongoing clinical trials or experimental therapies targeting sirtuins, would strengthen the review’s practical relevance.

Author’s reply to comment 4- Thank you for your valuable suggestion. We agree that expanding the discussion on translational aspects, including ongoing clinical trials and experimental therapies targeting sirtuins, would enhance the review’s practical relevance. However, we would like to highlight here that till date, sirtuins and their inhibitors have primarily only been investigated as modulators of signaling cascades in prostate cancer and none of these inhibitors advancing clinical trials. We certainly acknowledge reviewer point and recognize that that can be a great topic to discuss as potential for future research in this area. We have now expanded on this topic in the Conclusion and Perspectives section to provide a more comprehensive discussion (Highlighted section: line-1021-1026). Moreover, we have now included a new Table (Table 2 in the revised manuscript) that summarizes the specific roles of sirtuins in prostate tumorigenesis and identifies potential sirtuins as biomarkers for prostate cancer. (Highlighted section,lines-834). Future more, we have now also discussed possible limitations of sirtuin-based therapeutic approaches in prostate cancer (Highlighted section: lines: 1032-1041). We believe that these enhancements are intended to adequately address the reviewer’s suggestion and will strengthen the review’s practical relevance.

Comment 5: 4. Are the conclusions consistent with the evidence and arguments presented, and do they address the main question posed? Please also explain why this is/is not the case.

Yes, the conclusions are consistent with the evidence and arguments presented. The review successfully synthesizes the role of sirtuins in prostate cancer pathogenesis and aligns this understanding with the broader therapeutic potential of targeting these molecules. The main questions posed by the authors are adequately addressed through a logical and evidence-based discussion.

Author’s reply to comment 5- Thank you for your feedback. We’re glad the conclusions align with the evidence and that the review effectively addresses key questions on sirtuins' role in prostate cancer.

Comment 6: 5. Are the references appropriate?

The references are appropriate and relevant.

Author’s reply to comment 6- Thank you for your feedback.

Round 2

Reviewer 1 Report

Comments and Suggestions for Authors

Thank you to the authors for making an effort to revise the manuscript. The suggestion of dividing into two manuscripts was not accepted, but that is their decision. Approximately 7 pages were reduced, a new table added and the structural part of sirtuins shortened. Despite these efforts, the manuscript must be revised because some changes indicated in specific lines do not match. For example, the reviewer is referred to the Conclusion and Perspective section in the revised manuscript (Highlighted sections 1026-1029). Similarly, some lines suggest that changes in the role of sirtuins in regulating DNA repair do not match the sections where the authors state that changes have been made. .